# PLD: A Choice-Theoretic List-Wise Knowledge Distillation

**Ejafa Bassam**[*]   **Dawei Zhu**[†]   **Kaigui Bian**[‡]
School of Computer Science, Peking University

## Abstract

Knowledge distillation is a model compression technique in which a compact "student" network is trained to replicate the predictive behavior of a larger "teacher" network. In logit-based knowledge distillation, it has become the de facto approach to augment cross-entropy with a distillation term. Typically, this term is either a KL divergence that matches marginal probabilities or a correlation-based loss that captures intra- and inter-class relationships. In every case, it acts as an additional term to cross-entropy. This term has its own weight, which must be carefully tuned. In this paper, we adopt a choice-theoretic perspective and recast knowledge distillation under the Plackett–Luce model by interpreting teacher logits as "worth" scores. We introduce *Plackett-Luce Distillation (PLD)*, a weighted list-wise ranking loss. In PLD, the teacher model transfers knowledge of its full ranking of classes, weighting each ranked choice by its own confidence. PLD directly optimizes a single "teacher-optimal" ranking. The true label is placed first, followed by the remaining classes in descending teacher confidence. This process yields a convex and translation-invariant surrogate that subsumes weighted cross-entropy. Empirically, across CIFAR-100, ImageNet-1K, and MS-COCO, PLD achieves consistent gains across diverse architectures and distillation objectives, including divergence-based, correlation-based, and feature-based methods, in both homogeneous and heterogeneous teacher–student pairs.

## 1   Introduction

Deep neural networks (DNNs) have achieved remarkable success across a wide array of tasks-from image classification and object detection to semantic segmentation and beyond [47, 41]. Accuracy, generalization, and robustness usually improve as models become deeper and larger [42]. However, these gains require more computation and memory, which limits deployment on mobile or embedded devices.

Knowledge distillation (KD) [12] addresses this by training a compact "student" network to mimic a large "teacher" network. Classically, KD minimizes a weighted sum of cross-entropy on hard labels and a temperature-scaled Kullback-Leibler (KL) divergence between teacher and student logits. This simple framework enriches the student's learning signal and has seen widespread adoption for model compression and robustness enhancement [9, 34].

Despite its popularity, recent studies reveal a paradox. Distilling from larger or more accurate teachers can sometimes *degrade* the student's performance, especially when there is a capacity mismatch [24, 33, 13]. Several methods have been proposed to address this issue. These include teacher-assistant frameworks [24, 33], selective distillation [3, 54], and auxiliary classifiers [18]. However, these methods often add extra architectural components or training stages. An alternative line of

---

[*]Email: ejafabassam@stu.pku.edu.cn

[†]Email: dwzhu@pku.edu.cn

[‡]**Corresponding author**; Email: bkg@pku.edu.cn

39th Conference on Neural Information Processing Systems (NeurIPS 2025).

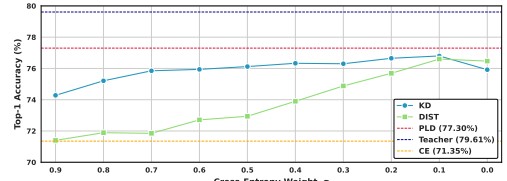 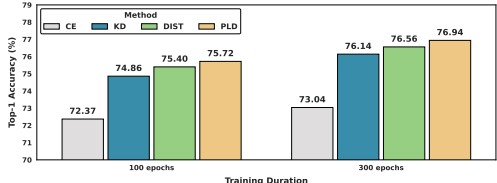

(a) CE-weight sensitivity for KD and DIST.   (b) KD, DIST, and PLD after 100 vs. 300 epochs.

Figure 1: **(a)** Varying the CE mixing weight $\alpha$ reveals that KD and DIST have different sensitivities-too much CE hurts both, while a sweet spot near $\alpha \approx 0.1$ maximizes Top-1 accuracy. **(b)** Under extended training (100 vs. 300 epochs), PLD consistently outperforms both KD and DIST, demonstrating its sustained gains.

work, DIST [13], remains within the classical KD pipeline yet replaces the KL-based distillation term. DIST argues that matching only marginal probabilities via KL divergence fails to preserve the relational structure encoded in the teacher's outputs.

Instead, DIST employs a Pearson correlation-based loss to preserve both inter-class correlations within each prediction and intra-class correlations across examples. Nevertheless, like other distillation methods, DIST still requires a separate cross-entropy term. As Figure 1a shows, reducing the weight on this cross-entropy term can improve accuracy, but removing it entirely leads to performance drop. Therefore, a unified objective that combines cross-entropy and distillation is desirable. Such an objective should derive its weights from the teacher's confidence rather than manual tuning.

To overcome these limitations, we propose a list-wise, choice-theoretic perspective on distillation. We interpret logits as "worth" scores under the classical Plackett–Luce model [22, 28]. We then derive a single *teacher-optimal* ranking, where the true label comes first and the remaining classes follow in descending teacher confidence. This ranking is imposed on the student using the *unsoftened* Plackett–Luce likelihood. The PL model is based on Luce's Choice Axiom, which ensures that choice probabilities are invariant to irrelevant alternatives. Plackett's extension generalizes this to full rankings, forming a coherent distribution over permutations through sequential removal. The first selection term of the PL model matches the cross-entropy on the true label. Thus, it naturally includes the standard KD cross-entropy loss. However, similar to [31, 16], we observe that proper weighting of each subsequent selection is crucial (see Sec. 5). We weight each step $k$ by the teacher's softmax mass $\alpha_k$. This leads to the *Plackett–Luce Distillation (PLD) Loss*, a convex and translation-invariant surrogate. PLD unifies cross-entropy, ListMLE [45], and P-ListMLE [16] as special cases. This choice-theoretic foundation captures the full ranking structure provided by the teacher and enables efficient, gradient-based optimization, eliminating the need for separate cross-entropy terms or manual weight adjustments.

Our main contributions are:

- We introduce a novel list-wise distillation objective that enforces the *teacher-optimal permutation* $\pi^*$-the unique ordering that places the ground-truth label first and ranks all other classes by descending teacher logits-via the Plackett-Luce likelihood, with each selection step $k$ weighted by the teacher's softmax mass $\alpha_k$, thereby eliminating the need for ad-hoc weight tuning.

- We show PLD is convex, smoothly differentiable with closed-form gradients, and subsumes CE, ListMLE, and P-ListMLE. Like classical KD and DIST, PLD is simple and efficient, easy to implement (see Appendix C), and requires no architectural modifications.

- We empirically demonstrate on CIFAR-100, ImageNet-1K, and MS-COCO that PLD achieves consistent gains across diverse architectures and distillation objectives, including divergence-based, correlation-based, and feature-based methods, in both homogeneous and heterogeneous teacher–student pairs.

The rest of the paper is organized as follows. Section 2 reviews prior work on KD and ranking losses. Section 3 presents necessary preliminaries. Section 4 derives the PLD loss and analyzes its properties. Section 5 reports empirical results. Finally, Section 6 concludes with a discussion of future directions.

## 2 Related Work

### 2.1 Knowledge Distillation and Capacity-Mismatch Remedies

Knowledge distillation (KD) [12], inspired by model-bootstrapping techniques [53], trains a compact student to mimic a larger teacher. It minimizes a temperature-scaled KL divergence between their output logits (softened probabilities). This process transfers both accuracy and robustness [9] while enabling model compression [34]. Over time, this simple yet effective framework has become a cornerstone for compressing and enhancing deep neural networks. However, a range of studies have shown that students distilled from very large or highly accurate teachers can underperform, even under adversarially robust settings [3, 26, 24, 33, 55, 39, 54, 13, 30, 18, 50, 48]. This counterintuitive degradation is mainly ascribed to a capacity mismatch between teacher and student. To address this issue, several architecture-level methods introduce intermediate or auxiliary models. Teacher Assistant KD (TAKD) employs a mid-sized assistant to bridge the gap [24]. Densely Guided KD (DGKD) aggregates multiple assistants for richer supervision [33]. Neighbor Self-KD (NSKD) adds auxiliary classifiers within student layers [18]. Student Customized KD (SCKD) adjusts the distillation loss based on gradient alignment with the student's objective [55]. Teacher-knowledge regularization techniques also refine the teacher's signal. Early-stopped teachers often outperform fully converged ones [3]. CheckpointKD selects intermediate checkpoints to prevent over-specialization [39]. Some methods exclude undistillable classes to focus supervision on reliable predictions [54]. Adapter modules control label smoothness [30], and Student-friendly KD (SKD) simplifies teacher outputs through a softening-simplifier pipeline [50].

### 2.2 Logit-Based Distillation Beyond KL Divergence

This trend is reflected in recent logit-based distillation methods that seek alternatives to KL divergence. Relational KD (RKD) [26] transfers mutual relations among data examples via distance and angle-wise losses that penalize structural discrepancies. DIST [13] introduced a Pearson correlation coefficient-based loss to explicitly capture the teacher's intrinsic inter and intra-class relations. Correlation Matching KD (CMKD) [25] reinterprets inter-class rankings in terms of the decision boundary and employs both Pearson and Spearman correlation losses, dynamically weighted by sample difficulty using a differentiable sorting operation. It was also observed [6] that better teacher calibration correlates with improved KD performance under standard methods. This finding motivated calibration-insensitive metrics such as ranking-based losses. These methods are computationally efficient, require no complex training pipelines or architectural overhead, yet achieve competitive performance. This observation motivates our search for a similarly efficient, calibration-insensitive objective grounded in list-wise ranking theory.

### 2.3 Ranking-Based and List-Wise Losses for Distillation

Early ranking methods focused on pairwise losses. In these methods, each pair of items is classified as correctly or incorrectly ordered, as in the ranking SVM [11, 14]. Listwise approaches, which operate over entire item lists, later gained traction in information retrieval and machine learning [2, 45]. Many draw on classical statistical ranking models-most notably Luce's choice model [22] and the Plackett-Luce permutation distribution [28]. As the number of items increases, even stochastic top-$k$ extensions such as ListNet [23] become impractical. Moreover, these losses are designed for direct supervision and do not naturally extend to knowledge distillation.

More recent work applies ranking losses to classification and distillation. For multiclass tasks, listwise losses have been proposed [40, 7], while distillation under a position-aware binary ranking loss has been studied [37, 8]. The method in [37] removes items that the teacher ranks low. In contrast, [17] adapts ranking-based distillation to collaborative filtering. Similarly, RankDistil [31] preserves the teacher's top-$k$ ordering by matching the student's item order to the teacher's and penalizing lower-ranked items.

Building on ListMLE [45], we directly optimize the likelihood of a single teacher-optimal permutation under the Plackett-Luce model. Inspired by RankDistil [31] and position-aware ListMLE (P-ListMLE) [16], we assign greater weight to top-ranked classes than to those ranked lower. Unlike RankDistil, which is designed for general ranking tasks and optimizes top-$k$ metrics, our method focuses on classification tasks. In this setting, top-1 accuracy is most important.

# 3 Preliminaries

We first review the multiclass classification and classical knowledge-distillation framework, then introduce the Plackett-Luce permutation model for rankings.

## 3.1 Multiclass Classification and Knowledge Distillation

Let $\mathcal{D} = \{(\mathbf{x}^{(n)}, y^{(n)})\}_{n=1}^N$ be a training set of $N$ examples, where each $\mathbf{x}^{(n)} \in \mathbb{R}^d$ and $y^{(n)} \in \{1, \ldots, C\}$. A neural network $f_\theta : \mathbb{R}^d \to \mathbb{R}^C$ maps an input $x$ to a logit vector $s = f_\theta(x) = (s_1, \ldots, s_C) \in \mathbb{R}^C$, which we refer to as the network's *logits* $s \in \mathbb{R}^C$. These logits induce a softmax distribution over classes:

$$p(y = i \mid s) = \frac{\exp(s_i)}{\sum_{j=1}^C \exp(s_j)}, \quad i = 1, \ldots, C.$$

The standard cross-entropy loss is

$$\mathcal{L}_{\mathrm{CE}}(s, y) = -\log p(y \mid s) = -s_y + \log \sum_{j=1}^C e^{s_j},$$

which depends only on the target logit $s_y$ and is therefore *intransitive*: any permutation of the other logits leaves $\mathcal{L}_{\mathrm{CE}}$ unchanged.

In knowledge distillation, a pretrained teacher $f^T$ and a student $f^S$ produce logits $t = f^T(x)$, $s = f^S(x)$.

We soften these via temperature $\tau > 0$:

$$q_i^T = \frac{\exp(t_i/\tau)}{\sum_{j=1}^C \exp(t_j/\tau)}, \qquad q_i^S = \frac{\exp(s_i/\tau)}{\sum_{j=1}^C \exp(s_j/\tau)},$$

and measure their divergence via a temperature-scaled KL term:

$$\mathcal{L}_{\mathrm{KD}}(s, t) = \tau^2 \, \mathrm{KL}(q^T \| q^S) = \tau^2 \sum_{i=1}^C q_i^T \log \frac{q_i^T}{q_i^S}.$$

The student minimizes the combined loss

$$\mathcal{L}(s, y) = \alpha \, \mathcal{L}_{\mathrm{CE}}(s, y) + (1 - \alpha) \, \mathcal{L}_{\mathrm{KD}}(s, t),$$

where $\alpha \in [0, 1]$ balances fitting the hard labels against matching the teacher's full output distribution. Minimizing $\mathcal{L}$ encourages the student both to place the correct class first and to mirror the teacher's full distribution. In Section 4, we extend this perspective by imposing a full Plackett-Luce ranking over the logits.

## 3.2 Plackett-Luce (PL) Ranking Model

Let $s \in \mathbb{R}^C$ be the logit vector produced by the network. We write $s_i$ for its $i$th component and interpret $i \succ j \iff s_i > s_j$ as "class $i$ preferred to class $j$." A full ranking is a permutation $\pi = (\pi_1, \ldots, \pi_C) \in S_C$. The Plackett-Luce model assigns each class $i$ a strictly positive worth $w_i = \phi(s_i) > 0$, where $\phi : \mathbb{R} \to \mathbb{R}_{>0}$ is strictly increasing (we take $\phi(s) = e^s$). It defines a probability over permutations by

$$P_{\mathrm{PL}}(\pi \mid s) = \prod_{k=1}^C \frac{w_{\pi_k}}{\sum_{l=k}^C w_{\pi_l}} = \prod_{k=1}^C \frac{\exp(s_{\pi_k})}{\sum_{l=k}^C \exp(s_{\pi_l})}.$$

At each step $k$, the class $\pi_k$ is chosen from the remaining items in proportion to its worth.

A key property is *translation-invariance*: adding $\delta$ to all logits (i.e. $s_i \mapsto s_i + \delta$) multiplies every $w_i$ by $e^\delta$, which cancels in the softmax-style ratios. By contrast, scaling logits by $a > 0$ (i.e. $s_i \mapsto a \, s_i$) changes the sharpness of the distribution.

The PL model factorizes the full $C$!-way ranking distribution into a chain of softmax terms. This factorization provides an efficient likelihood for any fixed permutation [45]. In Section 4, we fix a "teacher-optimal" permutation $\pi^*$ and minimize its negative log-likelihood under the student. This approach enforces both top-1 correctness and richer inter-class ordering information.

# 4 Method

We derive the PLD objective in two stages. First, we extract a *teacher-optimal permutation* from the teacher's logits using the Plackett–Luce model (Sec. 4.1). This step identifies the ranking target for distillation. Second, we introduce a *confidence-weighted likelihood* to form the final Plackett–Luce Distillation (PLD) loss (Sec. 4.2).

## 4.1 Teacher-Optimal Permutation under Plackett-Luce

From the Plackett-Luce ranking viewpoint, a classifier's logits define a sequential choice process among the $C$ classes. In the first selection step, the model chooses one class *out of all $C$* in proportion to $\exp(s_i)$, exactly matching the softmax probability used in standard cross-entropy. Consequently, minimizing cross-entropy enforces only the first-choice probability, i.e., the probability of selecting the correct class first. It remains *agnostic* to the ordering of the remaining $C - 1$ classes. In contrast, the full Plackett–Luce model assigns probabilities to all $C!$ possible permutations. It does this by chaining softmax-style selections at each step. Thus, cross-entropy can be seen as a *relaxed* Plackett-Luce loss that cares only about the top-1 selection.

If we augment cross-entropy with a knowledge-distillation term, such as KL matching or correlation-based loss, the objective still focuses mainly on the true class through the cross-entropy component. The distillation term only adjusts the remaining logits. Viewed through the Plackett–Luce perspective, this process enforces the first selection to be the ground-truth label, as in cross-entropy. It then shapes later choices according to the teacher's preferences instead of matching the entire distribution. Concretely, since Plackett-Luce makes each selection independently-choosing $\pi_k$ among the remaining classes in proportion to their worth-we fix the first pick to the true label and let the teacher's descending-logit ordering guide all later picks. Denoting the teacher's logit vector by $t = (t_1, \ldots, t_C) \in \mathbb{R}^C$, we refer to the resulting fixed ordering

$$\pi^* = \big(y, \ \text{argsort}(t) \setminus \{y\}\big)$$

as the *teacher-optimal permutation*, and use $\pi^*$ as our sole ranking target.

As shown in [45], one can construct surrogate losses on a single Plackett–Luce permutation. Examples include likelihood-based (ListMLE), cosine-similarity, and cross-entropy variants, all suitable for gradient-based optimization. Among these, the likelihood (ListMLE) surrogate stands out for its simplicity and for satisfying desirable properties such as consistency, soundness, continuity, differentiability, and convexity, while also exhibiting strong empirical performance. We adopt the ListMLE approach. We place an empirical one-hot target on the teacher-optimal permutation $\pi^*$ and define its negative log-likelihood under the student's logits as the unweighted loss.

$$\mathcal{L}_{\text{unweighted}}(s; \pi^*) = -\log P_{\text{PL}}\big(\pi^* \mid s\big) = -\sum_{k=1}^{C} \log \frac{\exp\big(s_{\pi_k^*}\big)}{\sum_{\ell=k}^{C} \exp\big(s_{\pi_\ell^*}\big)}.$$

This formulation naturally yields a one-hot target over the $C!$ Plackett-Luce permutations-assigning probability one to $\pi^*$ and zero to all others-and serves as the foundation for our weighted distillation objective.

## 4.2 Confidence-Weighted Likelihood Yielding PLD

In the standard PL model, each selection step is weighted equally-i.e., every position contributes identically to the likelihood. Position-aware ListMLE (P-ListMLE) [16] addresses this by introducing a fixed, strictly decreasing weight sequence, such as $\alpha_k = 2^{C-k} - 1$.

This scheme penalizes errors at top ranks more than those lower down. However, in multiclass classification-where only the top-1 decision ultimately matters-fixed, hand-crafted weight schedules are both hyperparameter-sensitive and fail to reflect the inherently greater importance of the first selection compared to subsequent ones. Observing that a pretrained teacher naturally assigns far greater confidence to the correct class than to the others, we therefore parameterize the weight at step $k$ by the teacher's softmax probability: $\alpha_k = q_{\pi_k^*}^T$

This choice ensures the loss automatically emphasizes the top-1 selection when the teacher is confident yet relaxes its focus when the teacher's output distribution is more uniform. Furthermore, our data-

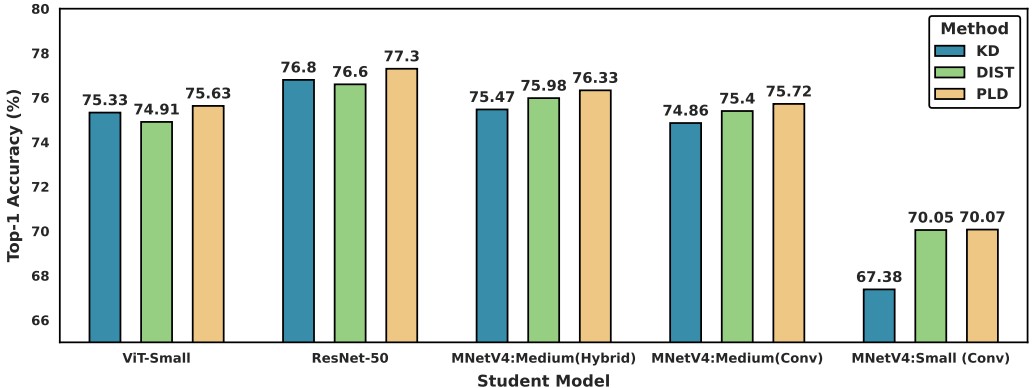

(a) Top-1 accuracy in the *homogeneous* distillation setting.

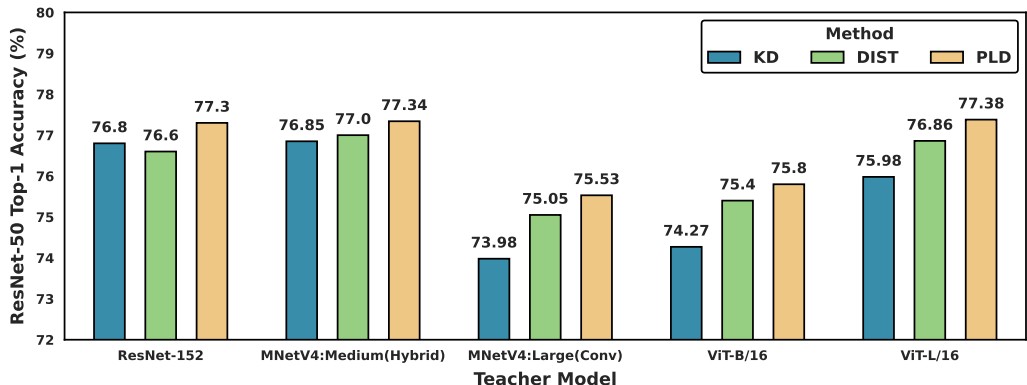

(b) Top-1 accuracy in the *heterogeneous* distillation setting.

Figure 2: **(a)** Homogeneous setting: larger teachers and smaller students within the same architecture family. **(b)** Heterogeneous setting: a fixed ResNet-50 student distilled from diverse teacher architectures.

driven weights recover several known ranking surrogates as special cases. Setting $\alpha = (1, 0, \dots, 0)$ gives the standard cross-entropy. A uniform $\alpha = (\frac{1}{C}, \dots, \frac{1}{C})$ produces a ListMLE-like objective, and any fixed decreasing sequence yields the P-ListMLE surrogate.

The resulting *Plackett-Luce Distillation (PLD) Loss* is defined as

$$\mathcal{L}_{\mathrm{PLD}}(s, t; y) = \sum_{k=1}^{C} q_{\pi_k^*}^{T} \left[ -s_{\pi_k^*} + \log \sum_{\ell=k}^{C} e^{s_{\pi_\ell^*}} \right],$$

thereby providing a unified, confidence-weighted ranking objective for knowledge distillation. Since PLD can be viewed as a teacher-softmax-weighted variant of ListMLE, it inherits the same favorable structural properties. In particular, each summand

$$-s_{\pi_k^*} + \log \sum_{\ell=k}^{C} e^{s_{\pi_\ell^*}}$$

is convex in the logits $s$, as it is the sum of an affine function and a log-sum-exp. Because nonnegative weighting by $q_{\pi_k^*}^{T}$ and summation over $k$ preserve convexity, the full PLD loss remains convex in $s$. This convexity, together with smooth differentiability, ensures efficient, gradient-based optimization. The gradient derivation is provided in Appendix A.

# 5 Experiments

We evaluate PLD on three representative visual recognition datasets: **CIFAR-100** [15], **ImageNet-1K** [4], and **MS-COCO** [20]. These datasets cover small- and large-scale image classification and general object detection.

We first benchmark PLD on **CIFAR-100** using traditional convolutional architectures and a unified training recipe. We then assess scalability on **ImageNet-1K**, where we adopt stronger architectures and optimized training strategies. Finally, we test PLD on **MS-COCO** for object detection to examine generalization beyond classification.

## 5.1 CIFAR-100 Classification

We compare **PLD** with classical and feature-based knowledge distillation methods on CIFAR-100 [15]. The dataset contains 50,000 training images and 10,000 validation images across 100 categories at $32 \times 32$ resolution. All models are trained from scratch for 250 epochs. We use the `AdamW` optimizer with $\beta_1 = 0.9$ and $\beta_2 = 0.999$. The learning rate follows a cosine-annealing schedule, starting at 0.001. We set weight decay to 0.5 and the batch size to 128.

Table 1: **Evaluation results on the CIFAR-100 dataset.** The upper and lower models denote the teacher and student, respectively.

| Method | Homogeneous setup | | | Heterogeneous setup |
|---|---|---|---|---|
| | WRN-40-2 WRN-40-1 | ResNet-56 ResNet-20 | ResNet-32×4 ResNet-8×4 | ResNet-50 MobileNetV2 |
| Student (CE) | 70.26±0.24 | 67.57±0.25 | 71.56±0.16 | 64.42±0.67 |
| *Feature-based methods* | | | | |
| AT [51] | 71.99±0.34 | 68.42±0.16 | 72.51±0.37 | 52.28±0.84 |
| FitNet [32] | 70.86±0.33 | 66.75±0.27 | 72.59±0.30 | 62.77±0.05 |
| PKT [27] | 71.43±0.16 | 68.66±0.20 | 72.86±0.17 | 66.08±0.11 |
| RKD [26] | 11.32±1.84 | 68.53±0.18 | 71.45±0.41 | 65.16±0.21 |
| SP [38] | 73.49±0.20 | 69.36±0.27 | 72.45±0.08 | 65.30±0.29 |
| VID [1] | 70.61±0.39 | 68.52±0.12 | 71.70±0.32 | 64.14±0.82 |
| *Logits-based methods* | | | | |
| KD [12] | 72.44±0.36 | 69.45±0.24 | 72.11±0.10 | 66.61±1.16 |
| DIST [13] | 72.74±0.33 | 69.66±0.49 | 73.21±0.10 | 66.98±0.74 |
| DKD [52] | 73.22±0.38 | 67.80±0.03 | 73.08±0.27 | **69.51±0.42** |
| PLD (ours) | **73.50±0.30** | **70.28±0.21** | **73.97±0.05** | 68.16±0.38 |

We apply standard data augmentations: random cropping, horizontal flipping, and per-channel normalization. We report the mean and standard deviation over three independent runs. For **DIST** [13], we set $\alpha = 1.0$ for the cross-entropy term, $\beta = 2.0$ for the inter-class relation loss, $\gamma = 2.0$ for the intra-class relation loss, and use a temperature of $\tau = 4.0$. **PLD** uses the same temperature ($\tau = 4.0$).

## 5.2 ImageNet-1K Classification

All experiments on ImageNet-1K [4] are conducted following the recipe [44] with minimal data augmentation. We train for 100 epochs with an effective batch size of 2048 images (256 per GPU across eight NVIDIA A100 SXM4 80 GB accelerators) using the LAMB[49] optimizer, an initial learning rate of $5 \times 10^{-3}$ decayed by a cosine schedule and linearly warmed up over the first 5 epochs, and weight decay fixed at 0.02. Under this setup, 100 epochs require approximately 6 hours for convolutional backbones and 8 hours for Vision Transformers. To isolate the effect of our losses, we use only random resized crops, horizontal flips, and per-channel normalization.

For validation we adopt the "A-recipe" from [44]. Specifically, we set the test resolution $r = 224$ and test crop ratio $\rho = 0.95$, then $\text{resize}_{\min} = \lceil r/\rho \rceil \approx 236$, apply a bicubic resize of the shorter side

to resize$_{\text{min}}$, followed by a center crop of size $r \times r$. We normalize by the same mean and standard deviation as in training.

Table 2: Top-1 accuracy (%) and student model size across student–teacher pairs on ImageNet-1K. "Teacher Acc" is the teacher's standalone Top-1; $\Delta_{\text{DIST}}$ and $\Delta_{\text{KD}}$ are PLD's gains over DIST and KD.

| Student | Teacher | Params(M) | Top-1 % Accuracy | | | | | |
| | | | Teacher | KD | DIST | PLD | $\Delta_{\text{DIST}}$ | $\Delta_{\text{KD}}$ |
|---|---|---|---|---|---|---|---|---|
| ViT-Small | ViT-Large(304.33M) | 22.05 | 84.80 | 75.33 | 74.91 | **75.63** | 0.72 | 0.30 |
| ResNet-50 | ResNet-152(60.19M) | 25.56 | 79.61 | 76.80 | 76.60 | **77.30** | 0.70 | 0.50 |
| **MobileNet-v4** | | | | | | | | |
| Hybrid-Medium | Large (conv) | 11.07 | 80.83 | 75.47 | 75.98 | **76.33** | 0.35 | 0.86 |
| Medium (conv) | (32.59M) | 9.72 | 80.83 | 74.86 | 75.40 | **75.72** | 0.32 | 0.86 |
| Small (conv) | | 3.77 | 80.83 | 67.38 | 70.05 | **70.07** | 0.02 | 2.69 |

Table 3: Top-1 % accuracy of ResNet-50 distilled from diverse teachers on ImageNet-1K. "Params" is the teacher's parameter count in millions. $\Delta_{\text{DIST}}$ and $\Delta_{\text{KD}}$ report PLD's gains over DIST and KD, respectively.

| Teacher | Params (M) | Top-1 % Accuracy | | | | $\Delta$ | |
| | | Teacher | KD | DIST | PLD | $\Delta_{\text{DIST}}$ | $\Delta_{\text{KD}}$ |
|---|---|---|---|---|---|---|---|
| ResNet-152 | 60.19 | 79.61 | 76.80 | 76.60 | **77.30** | 0.70 | 0.50 |
| MobileNet-v4 Hybrid Medium | 11.07 | 78.66 | 76.85 | 77.00 | **77.34** | 0.34 | 0.49 |
| MobileNet-v4 Conv Large | 32.59 | 80.83 | 73.98 | 75.05 | **75.53** | 0.48 | 1.55 |
| ViT-Base/16 | 86.57 | 82.07 | 74.27 | 75.40 | **75.80** | 0.40 | 1.53 |
| ViT-Large/16 | 304.33 | 84.80 | 75.98 | 76.86 | **77.38** | 0.52 | 1.40 |

Table 4: Top-1 accuracy (%) of the MobileNet-v4 Medium student distilled from a MobileNet-v4 Large teacher after 100 and 300 training epochs. The "$\Delta$" row shows the improvement from 100 to 300 epochs.

| Epochs | CE | KD | DIST | PLD | $\Delta_{\text{DIST}}$ | $\Delta_{\text{KD}}$ |
|---|---|---|---|---|---|---|
| 100 epochs | 72.37 | 74.86 | 75.40 | **75.72** | 0.32 | 0.86 |
| 300 epochs | 73.04 | 76.14 | 76.56 | **76.94** | 0.38 | 0.80 |
| $\Delta$ (300–100) | 0.67 | 1.28 | 1.16 | 1.22 | – | – |

#### 5.2.1 Baseline Configuration and Ablation

We evaluate four losses under identical training: CE, KD [12], DIST [13], and PLD. We use a ResNet-50 student and a ResNet-152 teacher [10]. Each objective replaces only the logit-based term and has comparable cost. Thus, KD (divergence-based) and DIST (correlation-based) are natural baselines for our ranking-based PLD. We sweep the key hyperparameters for each method: $\alpha$ for KD, $\beta, \gamma$ for DIST, and $\tau_T$ for PLD. Figure 1a shows the effect of the CE mixing weight. Reducing $\alpha$ improves KD and DIST up to a region near $\alpha \approx 0.1$. Setting $\alpha = 0$ (no CE) degrades performance. Table 5 reports results with and without CE (see Table 8 for the full ablation). Table 6 lists PLD-specific sweeps.

#### 5.2.2 Ablation of PLD

As PLD weights each position by $\alpha_k = \dfrac{\exp\left(t_{\pi_k^*}/\tau_T\right)}{\sum_{j=1}^C \exp\left(t_{\pi_j^*}/\tau_T\right)}$, we sweep $\tau_T \in \{0.5, 1.0, 1.5, 2.0, 4.0\}$ to assess sensitivity: lower $\tau_T$ sharpens the distribution, while higher $\tau_T$ flattens the distribution. We also test two special cases: **Uniform (ListMLE):** $\alpha_k = 1/C$. **Position-aware (p-ListMLE):** $\alpha_k = q_{\pi_k^*}^T / \sum_j q_{\pi_j^*}^T$.

As Table 6 shows, the unsoftened PLD ($\tau_T = 1$) consistently achieves the highest Top-1/Top-5 accuracies, with uniform ListMLE trailing by 2-3 pp. Overall, the best results occur at $\tau_T = 1.0$ (no softening) or with only slight perturbations ($1.0 \pm \epsilon$), demonstrating that PLD is robust to the exact choice of $\alpha$ distribution and does not require aggressive sharpening or over-softening.

Table 5: Top-1 and Top-5 accuracy for baseline methods (ResNet-50 student, ResNet-152 teacher with Top-1 Acc. $79.61\%$). "–" indicates not applicable. **CE:** standard cross-entropy. **LS:**[36] cross-entropy with label smoothing ($\epsilon = 0.1$). **KD:** vanilla distillation with mixing weight $\alpha$ and temperature $\tau = 2$. **DIST:** relational distillation with inter-class weight $\beta$, intra-class weight $\gamma$, and temperature $\tau = 1$.

| Method | $\alpha$ | $\beta$ | $\gamma$ | $\tau$ | Top-1 Acc.% | Top-5 Acc.% |
|---|---|---|---|---|---|---|
| Teacher (ResNet-152) | – | – | – | – | 79.61 | – |
| CE | – | – | – | – | 71.35 | – |
| LS ($\epsilon = 0.1$) | – | – | – | – | 73.92 | – |
| KD | 0.00 | – | – | 2.00 | 75.92 | 92.82 |
| | 0.10 | – | – | 2.00 | **76.80** | 93.16 |
| DIST | 0.00 | 0.50 | 0.50 | 1.00 | 76.47 | 93.19 |
| | 0.10 | 0.45 | 0.45 | 1.00 | **76.60** | 93.30 |

Table 6: Top-1 and Top-5 accuracy for PLD variants (ResNet-50 student, ResNet-152 teacher with Top-1 Acc. $79.61\%$). "–" indicates not applicable.

| Variant | $\tau$ | Top-1 Acc.% | Top-5 Acc.% |
|---|---|---|---|
| PLD | 0.50 | 76.06 | 92.39 |
| | 1.00 | **77.30** | 93.28 |
| | 1.50 | 77.17 | 92.74 |
| | 2.00 | 76.46 | 92.34 |
| | 4.00 | 75.22 | 91.34 |
| PLD–ListMLE | – | 74.11 | 90.60 |
| PLD–pListMLE | – | 76.60 | 93.23 |

### 5.2.3 Distillation Across Homogeneous Architectures

Figure 2(a) evaluates three backbone families in a homogeneous setting: ResNet [10], MobileNet-v4 [29], and Vision Transformers [5]. Across convolutional models, PLD outperforms KD by 0.5–2.7 pp and DIST by 0.2–0.7 pp. On ViT, PLD yields +0.72 pp over DIST and +0.30 pp over KD. Averaged over all pairs, PLD improves Top-1 accuracy by **+0.42** pp versus DIST and **+1.04** pp versus KD. Table 2 reports full Top-1 results for KD, DIST, and PLD.

### 5.2.4 Distillation Across Diverse Teachers

Logit-based distillation can use teachers from different architectures as long as the logits match the student's dimensionality. Figure 2(b) shows the heterogeneous setting. We fix the student to ResNet-50 ($\approx 25.56$M parameters) and distill from a range of convolutional and transformer teachers. Across all teachers, PLD consistently outperforms KD and DIST. Its $\Delta_{\mathrm{DIST}}$ gains range from +0.34 pp (MobileNet-v4 Hybrid) to +0.70 pp (ResNet-152), averaging **+0.48** pp. Its $\Delta_{\mathrm{KD}}$ gains range from +0.49 pp to +1.55 pp, averaging **+1.09** pp. As teacher capacity increases, KD's benefit declines and DIST shows a modest upward trend. PLD follows this trend and further widens the margin over DIST for larger teachers. The largest gap between DIST and PLD appears in the ResNet-50←ResNet-152 pairing. Table 3 reports each teacher's Top-1 accuracy, parameter count, and the student's Top-1 under KD, DIST, and PLD, along with $\Delta_{\mathrm{DIST}}$ and $\Delta_{\mathrm{KD}}$.

### 5.2.5 Consistency under Extended Training

Our homogeneous experiments (Sec. 5.2.3) show that KD sometimes beats DIST and vice versa. We ask whether PLD consistently surpasses both under longer training. We train a MobileNet-v4 Medium student from a MobileNet-v4 Large teacher for 100 and 300 epochs with identical hyperparameters.

Table 7: **Object detection performance on MS-COCO.** We distill Faster R-CNN detectors with FPN [19] using DKD [52] and PLD. All models are trained under identical schedules. Results are reported on `val2017`.

| Teacher → Student | Method | AP | $AP_{50}$ | $AP_{75}$ | $AP_s$ | $AP_m$ | $AP_l$ |
|---|---|---|---|---|---|---|---|
| ResNet-50 → MobileNetV2-FPN | DKD | 29.24 | 50.42 | 29.87 | 16.15 | 31.11 | 38.07 |
| ResNet-50 → MobileNetV2-FPN | PLD | 28.91 | 49.63 | **29.88** | 16.05 | 30.41 | **38.59** |
| ResNet-101 → ResNet-18-FPN | DKD | 32.11 | 53.43 | 33.90 | 18.46 | 34.43 | 41.50 |
| ResNet-101 → ResNet-18-FPN | PLD | **32.47** | **53.83** | **34.17** | 18.50 | **34.97** | **42.12** |
| ResNet-101 → ResNet-50-FPN | DKD | 36.54 | **58.57** | 39.46 | **21.74** | 39.80 | **47.31** |
| ResNet-101 → ResNet-50-FPN | PLD | **36.60** | 58.28 | **39.58** | 21.37 | 39.76 | 47.24 |

Table 4 reports Top-1 on ImageNet-1K, and Figure 1b provides a detailed comparison. At 100 epochs, PLD achieves 75.72%. This is +0.32 pp over DIST and +0.86 pp over KD. At 300 epochs, PLD reaches 76.94% (+0.38 pp over DIST; +0.80 pp over KD). The PLD gains with longer training are +1.22 pp, which is nearly twice the standard pretraining gains (+0.67 pp). This scaling is comparable across KD (+1.28 pp) and DIST (+1.16 pp). Thus, PLD preserves and slightly widens its advantage under extended training.

## 5.3 Object Detection on MS-COCO

PLD extends to tasks where the teacher and student share the same output logits, including object detection and semantic segmentation. We distill Faster R-CNN [19] detectors on MS-COCO [20] with standard FPNs. We evaluate two convolutional teacher–student pairs: ResNet-50 → MobileNetV2-FPN and ResNet-101 → {ResNet-18-FPN, ResNet-50-FPN}. All models use identical hyperparameters, and schedules. We compare PLD to the strong DKD baseline [52].

Table 7 shows that PLD achieves comparable or slightly better performance than DKD across multiple pairs. PLD improves AP and $AP_{75}$ in most settings while maintaining similar $AP_s$ for small objects. These results indicate that PLD transfers structured knowledge for dense prediction without modifying the detector.

## 6 Conclusion and Limitation

In this work, we introduced *Plackett–Luce Distillation* (PLD), a unified, choice-theoretic framework for logit-based knowledge distillation. Empirically, across CIFAR-100, ImageNet-1K, and MS-COCO, PLD achieves consistent gains across diverse architectures and distillation objectives, including divergence-based, correlation-based, and feature-based methods, in both homogeneous and heterogeneous teacher–student pairs. These results show that transferring the teacher's structured preferences yields stronger and more stable student performance than marginal- or correlation-based objectives.

PLD applies broadly to any setting where the student and teacher share the same logit dimensionality. However, it assumes fully aligned class vocabularies and does not yet support mismatched label sets or incremental class addition. PLD also requires sorting over the $C$ output logits to extract the teacher-optimal permutation, which has $O(C \log C)$ complexity per example. Although efficient in distributed implementations, this cost is higher than the $O(C)$ complexity of standard KD or DIST. Empirical runtime analysis (Appendix I) shows that the overhead is negligible for common architectures. As with other distillation methods, PLD's benefits depend on teacher confidence and may diminish when the teacher's softmax distribution is nearly uniform, as illustrated in the loss landscape analysis (Appendix J).

PLD's choice-theoretic foundation opens several directions for future work. First, adaptive or curriculum-driven weighting schemes could adjust the ranking loss based on sample difficulty. Second, extending PLD to other domains, such as sequence modeling, reinforcement learning, or multitask learning, may offer similar gains. We hope this work encourages further exploration of ranking-based objectives for principled and effective model compression.

## Acknowledgments and Disclosure of Funding

This work is partially sponsored by National Key R&D Program of China under Grant 2022YFB3104200, NSFC U24A20235 and 62032003.

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

# A  Gradient Derivation for PLD Loss

## A.1  Definitions

Let $s = (s_1, \ldots, s_C) \in \mathbb{R}^C$ be the student's logits. Let $\pi^* = (\pi_1^*, \ldots, \pi_C^*)$ be a given permutation of class indices, referred to as the *teacher-optimal permutation*. Let $\alpha_k$ for $k = 1, \ldots, C$ be scalar weights, constant with respect to $s$.

The PLD loss function is defined as:

$$\mathcal{L}(s) = \sum_{k=1}^{C} \alpha_k \left[ -s_{\pi_k^*} + \log \sum_{\ell=k}^{C} \exp(s_{\pi_\ell^*}) \right].$$

We can write $\mathcal{L}(s) = \sum_{k=1}^{C} L_k(s)$, where

$$L_k(s) = \alpha_k \left[ -s_{\pi_k^*} + \phi_k(s) \right],$$

and

$$\phi_k(s) = \log \sum_{\ell=k}^{C} \exp(s_{\pi_\ell^*}).$$

Our goal is to compute the gradient $\nabla_s \mathcal{L}(s)$, whose $i$-th component is $\frac{\partial \mathcal{L}}{\partial s_i}$. Due to the linearity of differentiation,

$$\frac{\partial \mathcal{L}}{\partial s_i} = \sum_{k=1}^{C} \frac{\partial L_k}{\partial s_i}.$$

We will first compute $\frac{\partial L_k}{\partial s_i}$.

## A.2  Derivative of $L_k(s)$:

We have $L_k(s) = -\alpha_k s_{\pi_k^*} + \alpha_k \phi_k(s)$. Let's differentiate each term with respect to $s_i$.

**1. Derivative of the affine term:**  The first term is $-\alpha_k s_{\pi_k^*}$.

$$\frac{\partial}{\partial s_i} \left( -\alpha_k s_{\pi_k^*} \right) = -\alpha_k \frac{\partial s_{\pi_k^*}}{\partial s_i}.$$

Since $s_{\pi_k^*}$ is the component of $s$ at index $\pi_k^*$, its derivative with respect to $s_i$ is 1 if $i = \pi_k^*$ and 0 otherwise. This can be written using the indicator function $\mathbf{1}\{i = \pi_k^*\}$.

$$\frac{\partial}{\partial s_i} \left( -\alpha_k s_{\pi_k^*} \right) = -\alpha_k \mathbf{1}\{i = \pi_k^*\}.$$

**2.  Derivative of the log-sum-exp term:**  The second term is $\alpha_k \phi_k(s)$, where $\phi_k(s) = \log \sum_{\ell=k}^{C} \exp(s_{\pi_\ell^*})$.

$$\frac{\partial}{\partial s_i} \left( \alpha_k \phi_k(s) \right) = \alpha_k \frac{\partial \phi_k}{\partial s_i}.$$

To compute $\frac{\partial \phi_k}{\partial s_i}$, let $X_k(s) = \sum_{\ell=k}^{C} \exp(s_{\pi_\ell^*})$. Then $\phi_k(s) = \log X_k(s)$. Using the chain rule, $\frac{\partial \phi_k}{\partial s_i} = \frac{1}{X_k(s)} \frac{\partial X_k(s)}{\partial s_i}$. Now, we compute $\frac{\partial X_k(s)}{\partial s_i}$:

$$\begin{aligned}
\frac{\partial X_k(s)}{\partial s_i} &= \frac{\partial}{\partial s_i} \left( \sum_{\ell=k}^{C} \exp(s_{\pi_\ell^*}) \right) \\
&= \sum_{\ell=k}^{C} \frac{\partial}{\partial s_i} (\exp(s_{\pi_\ell^*})) \\
&= \sum_{\ell=k}^{C} \left( \exp(s_{\pi_\ell^*}) \cdot \frac{\partial s_{\pi_\ell^*}}{\partial s_i} \right) \\
&= \sum_{\ell=k}^{C} \exp(s_{\pi_\ell^*}) \mathbf{1}\{i = \pi_\ell^*\}.
\end{aligned}$$

The term $\mathbf{1}\{i = \pi_\ell^*\}$ is non-zero (equal to 1) only if $i = \pi_\ell^*$. This occurs if the index $i$ is part of the set of indices $\{\pi_k^*, \pi_{k+1}^*, \ldots, \pi_C^*\}$. If $i$ is in this set, then exactly one $\ell$ in the sum (from $k$ to $C$) will satisfy $\pi_\ell^* = i$, and for that specific $\ell$, the term becomes $\exp(s_i)$. If $i$ is not in this set, all terms are zero. Thus,

$$\frac{\partial X_k(s)}{\partial s_i} = \begin{cases} \exp(s_i) & \text{if } i \in \{\pi_k^*, \ldots, \pi_C^*\} \\ 0 & \text{otherwise.} \end{cases}$$

So, $\frac{\partial \phi_k}{\partial s_i}$ becomes:

$$\frac{\partial \phi_k}{\partial s_i} = \frac{1}{\sum_{\ell=k}^{C} \exp(s_{\pi_\ell^*})} \cdot \begin{cases} \exp(s_i) & \text{if } i \in \{\pi_k^*, \ldots, \pi_C^*\} \\ 0 & \text{otherwise} \end{cases}.$$

Let us define $\sigma_k(i)$ as:

$$\sigma_k(i) := \begin{cases} \dfrac{\exp(s_i)}{\sum_{\ell=k}^{C} \exp(s_{\pi_\ell^*})} & \text{if } i \in \{\pi_k^*, \ldots, \pi_C^*\} \\ 0 & \text{otherwise.} \end{cases}$$

Then, $\frac{\partial \phi_k}{\partial s_i} = \sigma_k(i)$. And the derivative of the second term of $L_k(s)$ is:

$$\frac{\partial}{\partial s_i} \left( \alpha_k \phi_k(s) \right) = \alpha_k \sigma_k(i).$$

**3. Combining derivatives for $L_k(s)$.** Now, we combine the derivatives of the two parts of $L_k(s)$:

$$\begin{aligned} \frac{\partial L_k}{\partial s_i} &= \frac{\partial}{\partial s_i} \left( -\alpha_k s_{\pi_k^*} \right) + \frac{\partial}{\partial s_i} \left( \alpha_k \phi_k(s) \right) \\ &= -\alpha_k \mathbf{1}\{i = \pi_k^*\} + \alpha_k \sigma_k(i) \\ &= \alpha_k \left[ \sigma_k(i) - \mathbf{1}\{i = \pi_k^*\} \right]. \end{aligned}$$

## A.3  Final Gradient $\nabla_s \mathcal{L}(s)$:

The $i$-th component of the gradient of the total loss $\mathcal{L}(s)$ is:

$$\frac{\partial \mathcal{L}}{\partial s_i} = \sum_{k=1}^{C} \frac{\partial L_k}{\partial s_i} \qquad\qquad = \sum_{k=1}^{C} \alpha_k \left[ \sigma_k(i) - \mathbf{1}\{i = \pi_k^*\} \right].$$

We can separate this sum into two parts:

$$\frac{\partial \mathcal{L}}{\partial s_i} = \sum_{k=1}^{C} \alpha_k \sigma_k(i) - \sum_{k=1}^{C} \alpha_k \mathbf{1}\{i = \pi_k^*\}.$$

For the first part of this expression, $\sum_{k=1}^{C} \alpha_k \sigma_k(i)$, the term $\sigma_k(i)$ is non-zero if and only if $i \in \{\pi_k^*, \ldots, \pi_C^*\}$. Therefore, this sum can be written as $\sum_{k\,:\,i \in \{\pi_k^*, \ldots, \pi_C^*\}} \alpha_k \frac{\exp(s_i)}{\sum_{\ell=k}^{C} \exp(s_{\pi_\ell^*})}$. For the second part, $\sum_{k=1}^{C} \alpha_k \mathbf{1}\{i = \pi_k^*\}$, the indicator function $\mathbf{1}\{i = \pi_k^*\}$ is equal to 1 if and only if $\pi_k^* = i$. Since $\pi^*$ is a permutation, for any given $i$, there is precisely one value of $k$ for which this condition is met. Consequently, this sum simplifies to $\sum_{k\,:\,\pi_k^* = i} \alpha_k$, which effectively selects the single $\alpha_k$ corresponding to the position of $i$ in the permutation $\pi^*$.

Combining these, the $i$-th component of the gradient is:

$$\frac{\partial \mathcal{L}}{\partial s_i} = \sum_{k\,:\,i \in \{\pi_k^*, \ldots, \pi_C^*\}} \alpha_k \frac{\exp(s_i)}{\sum_{\ell=k}^{C} \exp(s_{\pi_\ell^*})} - \sum_{k\,:\,\pi_k^* = i} \alpha_k.$$

In vector form, let $e_{\pi_k^*}$ denote the standard basis vector that is 1 at index $\pi_k^*$ and 0 elsewhere. Let $\sigma_k$ be a vector whose $j$-th component is $\sigma_k(j)$. Then the gradient of $L_k(s)$ is $\nabla_s L_k(s) = \alpha_k(\sigma_k - e_{\pi_k^*})$. The total gradient is:

$$\nabla_s \mathcal{L}(s) = \sum_{k=1}^{C} \alpha_k(\sigma_k - e_{\pi_k^*}).$$

## A.4 Convexity and Smoothness:

Each term $L_k(s) = \alpha_k \left[ -s_{\pi_k^*} + \log \sum_{\ell=k}^{C} \exp(s_{\pi_\ell^*}) \right]$ is a sum of an affine function $(-\alpha_k s_{\pi_k^*})$ and a non-negatively weighted log-sum-exp function $(\alpha_k \phi_k(s))$. Affine functions are convex. The log-sum-exp function $\phi_k(s)$ is known to be convex. Assuming $\alpha_k \geq 0$ (which is true if they are derived from softmax probabilities, as in $q_{\pi_k^*}^T$), the term $\alpha_k \phi_k(s)$ is also convex. The sum of convex functions is convex, so each $L_k(s)$ is convex. The total loss $\mathcal{L}(s) = \sum_{k=1}^{C} L_k(s)$ is a sum of convex functions, and therefore, $\mathcal{L}(s)$ is convex in $s$.

Furthermore, affine functions are smooth (infinitely differentiable). The log-sum-exp function is also smooth. Since differentiation, non-negative weighting, and summation preserve smoothness (specifically, Lipschitz continuity of the gradient on any bounded domain), the full PLD loss $\mathcal{L}(s)$ has a Lipschitz-continuous gradient on any bounded domain, which is beneficial for gradient-based optimization methods.

# B   Full Baseline Ablation

Table 8: **Hyperparameter grid search on ImageNet-1K.** Each configuration uses a ResNet-50 student distilled from a ResNet-152 teacher (Top-1 Acc. $79.61\%$). "−" indicates not applicable. The best setting in each group is highlighted.

| Method | Hyperparameters | | | | Accuracy (%) | |
|---|---|---|---|---|---|---|
| | $\alpha$ | $\beta$ | $\gamma$ | $\tau$ | Top-1 | Top-5 |
| **Teacher** | – | – | – | – | 79.61 | – |
| **CE** | – | – | – | – | 71.35 | – |
| **LS** ($\epsilon=0.1$) | – | – | – | – | 73.92 | – |
| *KD hyperparameter sweep ($\tau=2$)* | | | | | | |
| KD | 0.00 | – | – | 2.00 | 75.92 | 92.82 |
| | 0.10 | – | – | 2.00 | **76.80** | **93.16** |
| | 0.20 | – | – | 2.00 | 76.65 | 93.11 |
| | 0.30 | – | – | 2.00 | 76.30 | 93.02 |
| | 0.40 | – | – | 2.00 | 76.33 | 93.10 |
| | 0.50 | – | – | 2.00 | 76.12 | 93.12 |
| | 0.60 | – | – | 2.00 | 75.94 | 92.79 |
| | 0.70 | – | – | 2.00 | 75.85 | 92.71 |
| | 0.80 | – | – | 2.00 | 75.21 | 92.39 |
| | 0.90 | – | – | 2.00 | 74.28 | 91.59 |
| *DIST hyperparameter sweep ($\tau=1$)* | | | | | | |
| DIST | 1.00 | 2.00 | 2.00 | 1.00 | 75.77 | 92.67 |
| | 0.00 | 0.50 | 0.50 | 1.00 | 76.47 | 93.19 |
| | 0.10 | 0.45 | 0.45 | 1.00 | **76.60** | **93.30** |
| | 0.20 | 0.40 | 0.40 | 1.00 | 75.69 | 92.63 |
| | 0.30 | 0.35 | 0.35 | 1.00 | 74.88 | 92.18 |
| | 0.40 | 0.30 | 0.30 | 1.00 | 73.90 | 91.35 |
| | 0.50 | 0.25 | 0.25 | 1.00 | 72.94 | 90.95 |
| | 0.60 | 0.20 | 0.20 | 1.00 | 72.71 | 90.58 |
| | 0.70 | 0.15 | 0.15 | 1.00 | 71.85 | 90.16 |
| | 0.80 | 0.10 | 0.10 | 1.00 | 71.89 | 90.16 |
| | 0.90 | 0.05 | 0.05 | 1.00 | 71.40 | 89.88 |

# C    Implementation of the PLD Loss

```python
import torch
import torch.nn.functional as F

def create_adjusted_ranking(flat_logits: torch.Tensor,
                            flat_labels: torch.LongTensor
                            ) -> torch.LongTensor:
    """
    For each example, sort logits ascending, remove the true label,
    and append it at the end so it occupies the last (top-1) position.
    """
    _, sorted_idx = torch.sort(flat_logits, dim=-1, descending=False)  # [N, V]
    mask = sorted_idx != flat_labels.unsqueeze(-1)                     # [N, V]
    V = flat_logits.size(-1)
    assert torch.all(mask.sum(dim=-1) == V - 1),

    "Must remove exactly one true label per row."
    sorted_excl = sorted_idx[mask].view(-1, V - 1)                    # [N, V-1]
    return torch.cat([sorted_excl, flat_labels.unsqueeze(-1)], dim=-1)  # [N, V]

def plackett_luce_loss(student_logits: torch.Tensor,
                       teacher_logits: torch.Tensor,
                       labels: torch.LongTensor,
                       temperature: float = 1.0
                       ) -> torch.Tensor:
    """
    Computes the PLD loss:
      L = sum_k alpha_k [ logsumexp_k - s_k ],   alpha_k = softmax_teacher[pi*_k].
    """
    flat_s, flat_t, flat_lbl = prepare_for_classification(
        student_logits, labels, teacher_logits
    )
    ranking = create_adjusted_ranking(flat_t, flat_lbl)               # [N, V]
    s_perm = torch.gather(flat_s, dim=-1, index=ranking)              # [N, V]
    t_perm = torch.gather(flat_t, dim=-1, index=ranking)              # [N, V]
    log_cumsum = torch.logcumsumexp(s_perm, dim=-1)                   # [N, V]
    per_pos_loss = log_cumsum - s_perm                               # [N, V]
    teacher_prob = F.softmax(t_perm / temperature, dim=-1)            # [N, V]
    weighted = per_pos_loss * teacher_prob                           # [N, V]
    return weighted.sum(dim=-1).mean()                               # scalar
```

# D    Additional Experimental Setup

For these supplementary experiments, we adopt a homogeneous teacher-student configuration drawn from the MobileNetV4 [29] family. Specifically, the student network is MobileNetV4 Convolutional Small, and the teacher network is MobileNetV4 Hybrid Large. To ensure that any observed differences arise solely from the choice of loss function, we use exactly the same training hyperparameters as in our main experiments, including learning-rate schedules, batch size, weight decay, and the data-augmentation pipeline.

To comprehensively evaluate the relative merits of Plackett-Luce Distillation (PLD) compared to DIST and KD, we conduct experiments across multiple optimizers, various divergence weightings, runtime and several logit-standardization schemes. We further quantify distributional alignment by measuring the KL divergence between student and teacher softmax outputs throughout training. Finally, to gain geometric insight into each loss, we visualize the loss landscapes of PLD, KD, and DIST.

## E  Optimizer Ablation Study

In our main experiments, we adopt the Lamb [49] optimizer as prescribed by the Timm [43] library. To evaluate the robustness of the Plackett-Luce Distillation (PLD) loss under different optimization schemes, we perform an ablation study in which we replace Lamb with three alternatives: AdamW [21], Adan [46], and AdaBelief [56]. Table 9 reports the Top-1 accuracy of the MobileNetV4-Small student on ImageNet-1K for each optimizer, under the KD, DIST, and PLD losses. PLD consistently outperforms both KD and DIST, achieving an average Top-1 gain of 0.825% over DIST and 2.21% over KD, with maximum improvements of 0.91% and 2.70%, respectively.

Table 9: Top-1 accuracy (%) of the MobileNetV4-Small student under different optimizers, comparing DIST, KD, and PLD. $\Delta_{\mathrm{DIST}} = \mathrm{PLD} - \mathrm{DIST}, \ \Delta_{\mathrm{KD}} = \mathrm{PLD} - \mathrm{KD}$.

| Optimizer | DIST | KD | PLD | $\Delta_{\mathrm{DIST}}$ | $\Delta_{\mathrm{KD}}$ |
|---|---|---|---|---|---|
| AdaBelief | 69.91 | 68.92 | **70.80** | **0.89** | **1.88** |
| AdamW | 69.94 | 68.89 | **70.85** | **0.91** | **1.96** |
| Adan | 70.07 | 68.52 | **70.82** | **0.75** | **2.30** |
| Lamb (base) | 70.41 | 68.46 | **71.16** | **0.75** | **2.70** |

## F  Logit Standardization Analysis

Logit standardization-centering or normalizing the teacher and student logits before applying the distillation loss-has been proposed [35] to improve optimization stability and distribution alignment. We evaluate four variants on the MobileNetV4-Small student (distilled from MobileNetV4-Hybrid-Large) using the Lamb optimizer. Specifically, we consider:

**DIST + std. logits**: standardize both teacher and student logits before computing the DIST loss; **KD + std. logits**: standardize both teacher and student logits before the KL-based KD loss; **PLD + std. logits**: standardize both teacher and student logits before the PLD loss; **PLD + std. teacher logits**: standardize only the teacher logits for the PLD softmax weighting.

Table 10 reports Top-1 accuracy on ImageNet-1K for each standardization variant, using the unstandardized PLD baseline of 71.16%. While standardizing logits yields modest gains for both DIST and KD, it does not benefit PLD: applying standardization to either both teacher and student logits or to the teacher logits alone leads to a slight degradation in PLD's performance.

Table 10: Effect of logit standardization on Top-1 accuracy (%). PLD baseline (no standardization): 71.16%. $\Delta = 71.16 - \mathrm{Acc}$.

| Method | Top-1 Acc. (%) | $\Delta$ |
|---|---|---|
| DIST + std. logits | 71.12 | 0.04 |
| KD + std. logits | 69.14 | 2.02 |
| PLD + std. logits | 70.66 | 0.50 |
| PLD + std. teacher logits | 70.81 | 0.35 |

## G  Classification Accuracy Across Divergences

In the main text we adopt the standard knowledge-distillation loss based on the forward Kullback-Leibler divergence:

$$\mathcal{L}_{\mathrm{KD}} = \alpha \, \mathcal{L}_{\mathrm{CE}}(z_s, y) + (1 - \alpha) \, D_{\mathrm{KL}}\big(\mathrm{softmax}(z_t/T) \, \| \, \mathrm{softmax}(z_s/T)\big),$$

with $\alpha = 0.1$ and $T = 2$. In addition to this forward KL term, we evaluate two alternatives: the reverse Kullback-Leibler divergence $D_{\mathrm{KL}}\big(\mathrm{softmax}(z_s/T) \, \| \, \mathrm{softmax}(z_t/T)\big)$ and the Jensen-Shannon divergence $D_{\mathrm{JS}}\big(\mathrm{softmax}(z_t/T), \, \mathrm{softmax}(z_s/T)\big)$. Table 11 reports Top-1 accuracy (%) using each divergence measure. Jensen-Shannon yields a modest gain over forward KL, while reverse KL performs comparably.

Table 11: Top-1 accuracy (%) of vanilla KD under different divergence measures. The PLD baseline (fixed across rows) is **71.16%**. $\Delta_{\mathrm{KD}} = \mathrm{PLD} - \mathrm{KD}$.

| Divergence | KD Top-1 | $\Delta_{\mathrm{KD}}$ |
|---|---|---|
| Forward KL | 68.46 | 2.70 |
| Reverse KL | 67.44 | 3.72 |
| Jensen-Shannon | 69.83 | 1.33 |

## H    Distribution Matching Analysis

We evaluate how well each distillation loss aligns the student's output distribution with the teacher's by measuring the KL divergence between their softmax outputs at the end of training. Table 12 reports these KL values for both homogeneous teacher-student pairs (same model family) and heterogeneous pairs (cross-architecture). Lower KL indicates tighter alignment. Although the PLD loss trains the student to respect the teacher's preference ordering, in the homogeneous setups PLD achieves better alignment than DIST. However, since the KD loss explicitly minimizes a KL term, it yields the lowest divergence overall, outperforming both DIST and PLD.

Table 12: KL divergence of student vs. teacher softmax outputs under DIST, KD, and PLD. Lower is better.

| Teacher | Student | DIST | KD | PLD |
|---|---|---|---|---|
| **Homogeneous** | | | | |
| MobileNetV4-Large | MobileNetV4-Medium | 0.67 | 0.55 | 0.60 |
| MobileNetV4-Large | MobileNetV4-Small | 0.95 | 0.85 | 0.83 |
| MobileNetV4-Large | MobileNetV4-Hybrid-Medium | 0.64 | 0.52 | 0.60 |
| ViT-Large/16 | ViT-Base/16 | 0.47 | 0.42 | 0.72 |
| ViT-Large/16 | ViT-Small/16 | 0.55 | 0.53 | 0.69 |
| **Heterogeneous** | | | | |
| MobileNetV4-Large | ResNet50 | 0.71 | 0.60 | 0.65 |
| MobileNetV4-Hybrid-Medium | ResNet50 | 1.27 | 0.27 | 1.50 |
| ViT-Base/16 | ResNet50 | 0.43 | 0.39 | 0.66 |
| ViT-Large/16 | ResNet50 | 0.46 | 0.44 | 0.59 |

## I    Runtime Analysis

We evaluate the computational efficiency of PLD compared with other distillation methods discussed in Section 5. PLD involves a single sorting operation to obtain the teacher-optimal permutation, similar to ListMLE [45]; therefore, its complexity is only $O(C \log C)$ far lower than the $O(C!)$ enumeration of all possible rankings. In practice, this adds negligible overhead: PLD trains at nearly the same speed as logit-based methods such as KD, DIST, and DKD. Table 13 reports the total training time (in minutes) for all CIFAR-100 experiments in Table 1. Despite its list-wise formulation, PLD's runtime remains comparable to KD and DIST, and substantially faster than other methods such as RKD or VID.

## J    Loss Landscape Visualization

To gain geometric insight into the optimization landscapes induced by our three distillation losses (DIST, KD, PLD), we plot 3D surfaces and 2D contours over a two-dimensional slice of the student-logit space.

**Setup:**    Let $t \in \mathbb{R}^V$ be a random teacher-logit vector normalized to unit norm, and let $d_1, d_2 \in \mathbb{R}^V$ be two random orthonormal directions. For a grid of $(\alpha, \beta) \in [-5\|t\|, 5\|t\|]^2$, we define student logits $s(\alpha, \beta) = t + \alpha\, d_1 + \beta\, d_2$ and compute each loss $L(s(\alpha, \beta), t)$ at every grid point. Figure 4 plots the loss landscapes for DIST [13], KD [12], and **PLD (ours)**. While KD and PLD both induce

Table 13: **Total training time (minutes)** for all CIFAR-100 experiments. PLD achieves comparable efficiency to KD, DIST, and DKD despite its list-wise formulation.

| Model (Teacher Student) | CE | PLD | KD | DIST | DKD | AT | FitNet | PKT | RKD | SP | VID |
|---|---|---|---|---|---|---|---|---|---|---|---|
| ResNet32×4 ResNet8×4 | 32.06 | 62.84 | 62.18 | 62.24 | 62.46 | 65.39 | 63.96 | 62.53 | 68.36 | 62.73 | 78.40 |
| ResNet50 MobileNetV2 | 23.62 | 102.59 | 102.23 | 102.49 | 102.54 | 108.17 | 113.18 | 102.28 | 128.65 | 102.78 | 132.06 |
| ResNet56 ResNet20 | 22.97 | 33.42 | 33.57 | 33.81 | 33.46 | 34.55 | 33.30 | 33.89 | 36.21 | 33.67 | 36.57 |
| WideResNet-40-2 WideResNet-40-1 | 27.67 | 33.10 | 37.56 | 36.73 | 34.27 | 36.77 | 38.05 | 36.84 | 37.62 | 37.01 | 37.35 |

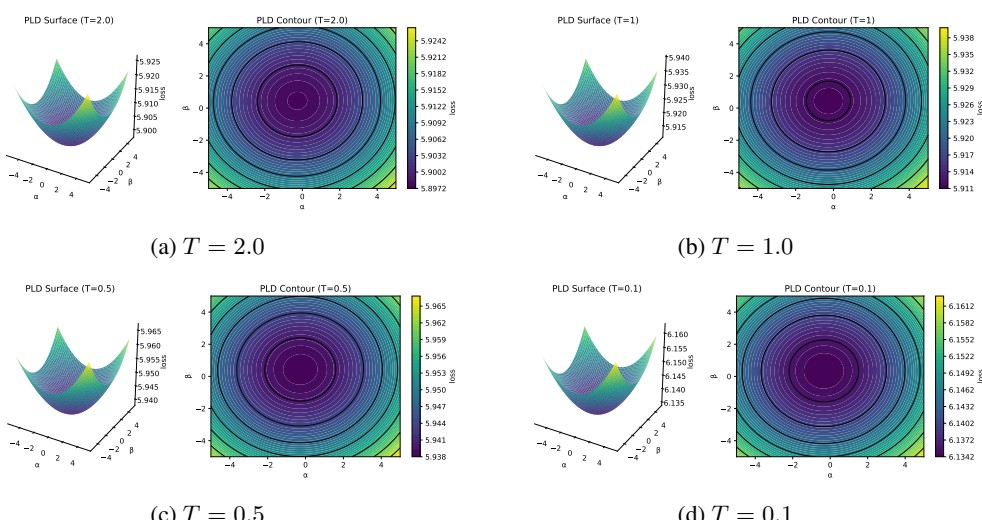

(a) $T = 2.0$   (b) $T = 1.0$

(c) $T = 0.5$   (d) $T = 0.1$

Figure 3: PLD loss surfaces at different teacher temperatures. (Top row) $T = 2.0$ and $T = 1.0$; (Bottom row) $T = 0.5$ and $T = 0.1$. Lowering $T$ below 1.0 flattens convexity.

convex surfaces, DIST dips sharply yet remains effectively planar in the $(\alpha, \beta)$ slice. Moreover, the contour for PLD is more tightly centered around the origin than that of KD.

## J.1 Temperature Sensitivity of the PLD Loss Surface

To investigate the effect of the teacher-softmax temperature $T$ on the geometry of the PLD loss landscape, we fix the same two-dimensional $(\alpha, \beta)$ slice and compute the PLD surface at four representative temperatures: $T \in \{2.0, 1.0, 0.5, 0.1\}$. Figure 3 shows both the 3D surface and 2D contour plots for each $T$. We observe that reducing $T$ from 2.0 down to 1.0 produces only minor changes, whereas further lowering $T$ below 1.0 flattens the curvature.

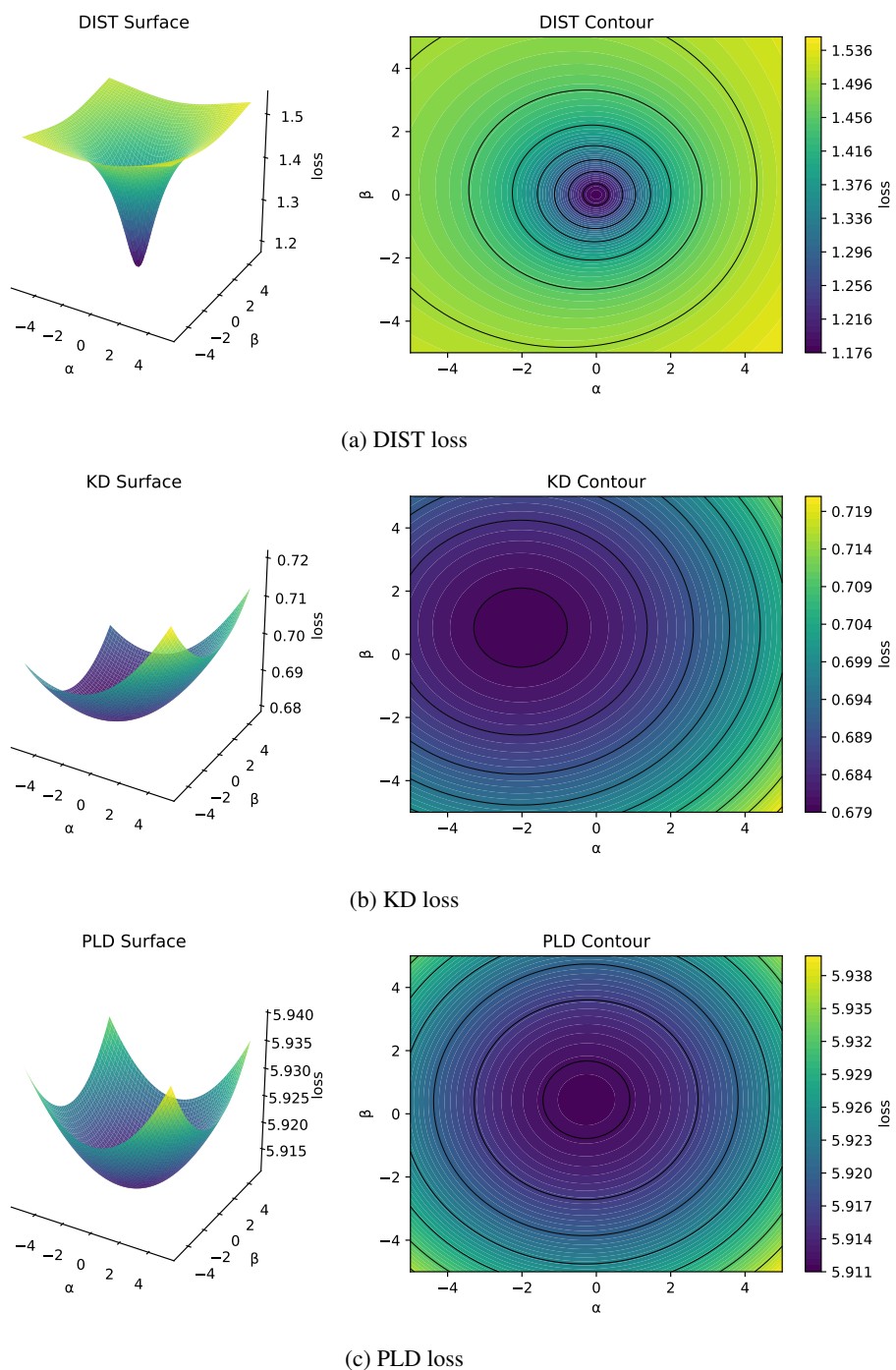

Figure 4: Loss landscapes of three distillation methods: (a) DIST exhibits a sharp dip yet remains effectively planar; (b) KD shows moderate convexity; (c) PLD (ours) exhibits better convexity with contours mostly centered at the origin.

