# OpenReview forum: "PLD: A Choice-Theoretic List-Wise Knowledge Distillation"
_NeurIPS.cc/2025/Conference — NeurIPS 2025 poster_

### Official Review · Reviewer_x7Lh · 2025-06-09

**Clarity:** 3
**Significance:** 3
**Originality:** 3
**Rating:** 4
**Confidence:** 4

**Summary:**

The authors propose a general framework for logit-based knowledge distillation from a choice-theoretic perspective. They introduce Plackett-Luce Distillation (PLD), a weighted list-wise ranking loss that interprets logits as "worth" scores. The proposed method outperforms standard logit-based baselines such as KD and DIST on standard image classification benchmarks.

**Questions:**

1. The writing can be improved for clarity and polish. In particular, there is an overuse of em dashes, which disrupts the flow of sentences. Additionally, lines 186 and 187 show inconsistent naming of ListMLE (referred to as both “likelihood” and “maximum-likelihood”), which should be made consistent.

2. The comparison is restricted to only two distillation methods (KD and DIST), whereas many other logit-based distillation methods exist. A broader comparison would strengthen the empirical claims.

3. In the Introduction, the primary motivation is that the Plackett-Luce model can unify cross-entropy and distillation-derived weights, unlike other methods which require an explicit cross-entropy term. However, it is not convincingly argued why having a separate cross-entropy term is a significant drawback in practice. Also, the theoretical benefit of adopting a list-wise ranking loss for image classification tasks remains unclear. what advantage does PLD offer over traditional approaches in this specific context?

4. In PLD, the logits from the teacher model are scaled by a temperature parameter \tau, but the student model's logits are not. This is inconsistent with the formulation used in temperature-scaled KL divergence, where both teacher and student logits are scaled. Could the authors explain the rationale behind this asymmetric design choice?

**Ethical Concerns:**

["NO or VERY MINOR ethics concerns only"]

**Final Justification:**

The author's rebuttal addressed my most of the concerns. Thus I will raise my score to boardline accept.

**Limitations:**

yes

**Quality:**

3

**Strengths And Weaknesses:**

Strengths:

1. The paper presents an interesting and novel direction by framing logit-based distillation through the lens of choice theory, offering a unified perspective on distillation losses.

2. The proposed PLD method achieves superior performance compared to existing logit-based distillation approaches (e.g., KD and DIST) across standard benchmarks.

Weaknesses:

1. Writing needs further improvement.

2. Too few comparison methods.

3. The motivation is not very clear.

---

> ### Author Rebuttal · Authors · 2025-07-31
>
> * We sincerely thank Reviewers fk7Y, Bcg7, ySdm, and x7Lh, as well as the Area and Senior Area Chairs, for their thorough and constructive feedback on PLD.
> * We are pleased that the reviewers recognize our method’s strong theoretical foundation (fk7Y, Bcg7), its efficiency and practical significance (fk7Y), the clarity of our derivations and writing (Bcg7, ySdm), and the robustness of our empirical evaluation (Bcg7, ySdm).
> * We greatly appreciate the suggestions to broaden our baseline comparisons, quantify runtime overhead, extend experiments to other tasks, and polish the manuscript’s readability. We will incorporate all additional experiments (CIFAR-100 baselines, COCO detection and runtime comparisons) into the revised manuscript.
> * We also thank the reviewers for highlighting minor typos and stylistic improvements, which we have addressed in the updated draft.
>
> Thank you Reviewer x7Lh for the detailed and constructive feedback. We are glad you found PLD a novel and effective direction compared to standard logit-based baselines. Below we (1) address writing clarity and consistency by reducing overused em dashes and standardizing the terminology around ListMLE (now consistently described as the likelihood loss); (2) expand empirical comparisons with additional KD baselines on CIFAR-100 under controlled hyperparameters and with an object detection task; (3) clarify the motivation by explaining the practical drawbacks of a separate CE term and the benefit of list-wise ranking in image classification; and (4) clarify the asymmetric temperature design. We hope these additions and clarifications address your concerns, and we would be grateful for your reconsideration of the score.
>
> **Q1: Writing clarity & consistency**
> We have carefully revised the manuscript to improve readability and ensure consistent terminology. All over‑long, dash‑connected sentences have been broken into shorter, standalone sentences, and we’ve replaced excess em dashes with commas or parentheses for smoother flow. We also updated lines 186–187 to standardize every occurrence of ListMLE—short for *Listwise Maximum Likelihood Estimation*—to “ListMLE (likelihood loss),” removing the term “maximum‑likelihood” to match the original paper’s terminology [38].
>
> **Reference**
> [38] F. Xia, T.-Y. Liu, J. Wang, W. Zhang, & H. Li. *Listwise approach to learning to rank: theory and algorithm.* ICML pages 1192–1199, 2008.
>
> **Q2. Inclusion of additional KD baselines and tasks: CIFAR‑100 Baselines and Object Detection**
>
> Answer: We included DIST (NeurIPS 2022) as a baseline and structured our ImageNet-1K experiments to compare models horizontally across architectures and scales; to further strengthen the empirical comparison we added CIFAR-100 results against other recent KD methods, all trained under the same controlled hyperparameters: AdamW (β₁=0.9, β₂=0.999), learning rate $=0.001$, weight decay $=0.5$, batch size $=128$, for 250 epochs with a cosine-annealing schedule and a 10-epoch linear warm-up. Results are reported as mean $\pm$ std over three runs. Following prior work (e.g., KD and DIST) we set the temperature $\tau=4$ where applicable.
>
> **Table A: CIFAR‑100 performance**
>
> | Model (Teacher→Student) | CE | PLD | KD | DIST | DKD | AT | FitNet | PKT | RKD | SP | VID |
> | --- | --- | --- | --- | --- | --- | --- | --- | --- | --- | --- | --- |
> | ResNet32x4ResNet8x4 | 71.56 ± 0.16 | **73.97 ± 0.05** | 72.11 ± 0.10 | 73.21 ± 0.10 | 73.08 ± 0.27 | 72.51 ± 0.37 | 72.59 ± 0.30 | 72.86 ± 0.17 | 71.45 ± 0.41 | 72.45 ± 0.08 | 71.70 ± 0.32 |
> | ResNet50 MobileNetV2 | 64.42 ± 0.67 | 68.16 ± 0.38 | 66.61 ± 1.16 | 66.98 ± 0.74 | **69.51 ± 0.42** | 52.28 ± 0.84 | 62.77 ± 0.05 | 66.08 ± 0.11 | 65.16 ± 0.21 | 65.30 ± 0.29 | 64.14 ± 0.82 |
> | resnet56resnet20 | 67.57 ± 0.25 | **70.28 ± 0.21** | 69.45 ± 0.24 | 69.66 ± 0.49 | 67.80 ± 0.03 | 68.42 ± 0.16 | 66.75 ± 0.27 | 68.66 ± 0.20 | 68.53 ± 0.18 | 69.36 ± 0.27 | 68.52 ± 0.12 |
> | WideResNet‑40‑2WideResNet‑40‑1 | 70.26 ± 0.24 | **73.50 ± 0.3** | 72.44 ± 0.36 | 72.74 ± 0.33 | 73.22 ± 0.38 | 71.99 ± 0.34 | 70.86 ± 0.33 | 71.43 ± 0.16 | 11.32 ± 1.84 | 73.49 ± 0.20 | 70.61 ± 0.39 |
>
> As discussed in our Conclusion (Sec 6), PLD naturally extends to any task where student and teacher share the same output logits, including detection or segmentation. To validate this, we distilled Faster R‑CNN detectors on COCO for two teacher→student pairs using DKD  and PLD:
>
> **Table B: Object detection performance (COCO)**
>
> | Teacher → Student | Method | AP | AP₅₀ | AP₇₅ | AP_s | AP_m | AP_l |
> | --- | --- | --- | --- | --- | --- | --- | --- |
> | ResNet‑50 → MobileNetV2‑FPN | DKD | **29.24** | **50.42** | 29.87 | **16.15** | **31.11** | 38.07 |
> | ResNet‑50 → MobileNetV2‑FPN | PLD | 28.91 | 49.63 | **29.88** | 16.05 | 30.41 | **38.59** |
> | ResNet‑101 → ResNet‑18‑FPN | DKD | 32.11 | 53.43 | 33.9 | 18.46 | 34.43 | 41.5 |
> | ResNet‑101 → ResNet‑18‑FPN | PLD | **32.47** | **53.83** | **34.17** | **18.5** | **34.97** | **42.12** |
> | ResNet‑101 → ResNet‑50‑FPN | DKD | 36.54 | **58.57** | 39.46 | **21.74** | **39.8** | **47.31** |
> | ResNet‑101 → ResNet‑50‑FPN | PLD | **36.6** | 58.28 | **39.58** | 21.37 | 39.76 | 47.24 |
>
> **Q3.1: why having a separate cross-entropy term is a significant drawback in practice?**
>
> Answer: The separate CE mixing weight is brittle, expensive to tune, and cannot adapt per sample; PLD subsumes CE into a single data-dependent ranking loss, eliminating that hyperparameter.
>
> Standard KD minimizes$$\mathcal{L} = \alpha\\mathrm{CE}(y,s) + (1-\alpha)\\mathrm{KL}(q^T\parallel q^S),$$
> which requires tuning both the CE weight $\alpha$ and the distillation weight $1-\alpha$. As Figure 1a shows (Sec. 1), small changes in $\alpha$ can shift Top-1 accuracy by more than 1%, and setting $\alpha=0$ (dropping CE entirely) causes a notable performance drop. This global hyperparameter is brittle, it doubles the tuning burden compared to CE alone, and it forces costly grid searches (Appendix B), especially because variants like DIST introduce additional weights such as temperature $\tau$ and balancing terms $\beta,\gamma$.
>
> Moreover, individual teacher output distributions vary per example: highly skewed logits require stronger softening to reveal inter-class structure, while flatter logits require less. A fixed $\alpha$ cannot adapt to this heterogeneity. In contrast, PLD’s first Plackett–Luce factor coincides with CE on the true class, and each subsequent ranking step is weighted by the teacher’s own softmax mass $\alpha_k = q^T_{\pi^*_k}$, making the supervision inherently data-dependent (Sec. 4.2). This unification subsumes CE into a single convex ranking loss and reduces the total hyperparameter count by one (removing $\alpha$).
>
> Motivated by Sutton’s “bitter lesson” that general methods scale best with compute, PLD offers a unified, self-tuning objective that reduces tuning overhead and focuses compute on scale rather than hyperparameter search.
>
> **Q3.2: What’s the advantage of list-wise ranking for image classification?**
>
> Answer: During inference, classification sorts the output logits—for example, it computes $\arg\max_i s_i$ or selects the top-$k$ values. Training typically uses the cross-entropy loss$$\mathcal{L}_{\mathrm{CE}} = -\sum_i y_i \log s_i.$$
> This loss enforces $s_y > s_i$ for all $i \neq y$. As shown in Section 4.1 (L172–L173), this constraint matches the first selection step in the Plackett–Luce model. Softmax guarantees $s_i > 0$ for every class, and cross-entropy training avoids overconfidence, yielding smooth output distributions. As a result, the detailed ordering among non-target classes is not fully exploited. In this sense, cross-entropy behaves like a relaxed ranking loss.Knowledge distillation captures these residual relationships via the teacher’s soft distribution $q^T$. For example, classes $a,b,c,d,e$ with true label $e$, cross-entropy enforces four independent constraints: $s_e > s_a$, $s_e > s_b$, $s_e > s_c$, and $s_e > s_d$. A list-wise ranking loss instead enforces the full ordering$$e > a > b > c > d
> $$in a single objective. This objective also enforces sub-rankings such as $a > b > c$ and $b > c > d$.Empirically, Table 2 shows that PLD’s ListMLE surrogate outperforms cross-entropy. This confirms that supervising the entire teacher-optimal permutation yields stronger Top-1 accuracy gains. Similar benefits of list-wise losses over pairwise losses have been observed in learning-to-rank methods such as ListNet \[1].
>
> **Reference**
>
> \[1] *Learning to rank: from pairwise approach to listwise approach.* ICML 2007.
>
>
> **Q4: Why is temperature applied only to the teacher logits in PLD, rather than symmetrically to both teacher and student as in temperature-scaled KL divergence?**
>
> **Answer:**  Applying a temperature $\tau$ to the student logits $s \mapsto s' = s/\tau$ simply rescales the entire PLD loss by $1/\tau$. Specifically,$$\mathcal{L}(s') = -\sum_k \alpha_k \log P_{\mathrm{student}}(\pi^*; s/\tau)= \frac{1}{\tau}\,\mathcal{L}(s).
> $$Under gradient descent with learning rate $\eta$, the update$$s \leftarrow s - \eta\,\frac{\partial \mathcal{L}(s')}{\partial s}
> = s - \frac{\eta}{\tau}\,\frac{\partial \mathcal{L}(s)}{\partial s}$$is equivalent to using the original loss with an effective learning rate $\eta' = \eta/\tau$. Thus student-side softening does not add modeling expressivity; it only changes the step size. In Supplementary Sec. 3 (Table 2) We find that standardizing both teacher and student logits yields 70.66% Top-1 accuracy, while standardizing only the teacher logits gives 70.81%, compared to 71.16% with no logit scaling.
>
> **Baselines & References**
> 1. **KD**:arXiv:1503.02531
> 2. **DIST** arXiv:2205.10536
> 3. **DKD**: arXiv:2203.08679
> 4. **AT**: arXiv:1612.03928
> 5. **FitNet**:arXiv:1412.6550
> 6. **PKT**:arXiv:1803.10837
> 7. **RKD**:  arXiv:1904.05068
> 8. **SP**:     arXiv:1907.09682
> 9. **VID**:    arXiv:1904.05835

---

> > ### Author Response · Authors · 2025-08-02
> >
> > **To Reviewer X7Lh:**
> > Due to space constraints and typos in our earlier answer to Q4, we’d like to clarify our response. Please treat the following as the official answer to that question:
> >
> > **Q4: Why apply temperature asymmetrically only to the teacher logits, not symmetrically to both teacher and student?**
> >
> > **Answer:** In PLD the asymmetry is intentional because temperature on the teacher side reshapes the *ranking weights* in a way that student-side scaling (i.e., $\mathbf{s}\mapsto \mathbf{s}/\tau$) does not. Let $s' := s/\tau$; then the temperature-scaled loss is $L_t(s):=L(s')$. By the chain rule,
> >
> > $$
> > \nabla_s L_t(s)
> > = \frac{1}{\tau}\nabla_{s'}L(s'),
> > $$
> >
> > so an update in $s$-space with step size $\eta_s$ corresponds to an implicit update in $s'$-space of size $\eta'=\eta_s/\tau^2$. Keeping the effective step in $s'$-space constant therefore requires adjusting $\eta_s=\tau^2\eta'$. In other words, student-side temperature primarily induces a learning-rate style scaling (with the accompanying softening of the student distribution that alters gradient direction), but it does *not* introduce the structured, per-transition modulation that PLD exploits.
> >
> > By contrast, PLD applies temperature to the **teacher** logits to modulate each rank transition in the ListMLE-style decomposition. For a target ordering like $e>a>b>c>d$, standard ListMLE treats all successive transitions equally, whereas PLD uses $\alpha_k$ to reflect the teacher’s confidence at each step. This both enforces the full ordering (since $\alpha_k>0$) and emphasizes stronger distinctions (e.g., $e>a$) over weaker ones (e.g., $c>d$). The temperature $\tau$ controls the concentration: $\tau>1$ flattens the $\alpha$s, spreading weight more evenly, while $\tau<1$ sharpens focus on top-ranked differences.
> >
> > Empirically, we take a data-centric approach in Supplementary Sec. 3 (Table 2) by standardizing the teacher logits—that is, setting $\tau=\operatorname{std}(t)$. Scaling only the teacher logits yields better Top-1 accuracy (70.81%) than scaling both teacher and student (70.66%) and comes close to the unscaled baseline (71.16%).
> >
> > We further verify this on CIFAR-100 by comparing the standard asymmetric PLD ($\text{teacher }\tau=4,\ \text{student }\tau=1$) to a symmetric variant where both teacher and student use $\tau=4$. The symmetric choice typically underperforms the asymmetric one:
> >
> > | Model (Teacher→Student)           | CE           | PLD (asymmetric) | PLD (symmetric) |
> > | --------------------------------- | ------------ | ---------------- | --------------- |
> > | ResNet32x4 → ResNet8x4            | 71.56 ± 0.16 | **73.97 ± 0.05** | 73.79           |
> > | ResNet50 → MobileNetV2            | 64.42 ± 0.67 | **68.16 ± 0.38** | 66.98           |
> > | ResNet56 → ResNet20               | 67.57 ± 0.25 | **70.28 ± 0.21** | 69.57           |
> > | WideResNet-40-2 → WideResNet-40-1 | 70.26 ± 0.24 | **73.50 ± 0.30** | 72.68           |

---

> > ### Comment · Reviewer_x7Lh · 2025-08-05
> >
> > Thanks for the author's rebuttal, which addressed my most of the concerns.
> > Thus I will raise my score to boardline accept.

---

> > > ### Author Response · Authors · 2025-08-05
> > >
> > > Thank you for your thoughtful critique and willingness to reconsider your score. Your feedback has helped strengthen our work.

---

### Official Review · Reviewer_ySdm · 2025-06-20

**Clarity:** 3
**Significance:** 3
**Originality:** 3
**Rating:** 5
**Confidence:** 3

**Summary:**

This paper targets logit distillation, which is knowledge distillation with merely logit outputs.
Previous logit distillation methods all rely on cross-entropy loss.
In this paper, they introduce a convex, translation-invariant surrogate, Plackett-Luce Distillation (PLD) Loss.
PLD views logits as “worth” scores in the classical Plackett-Luce permutation model.
With the designed mechanism, PLD can not only act as the cross-entropy loss but also capture the full rank structure provided by the teacher model.
Experiment results on the classification task prove the effectiveness of the proposed method.

**Questions:**

See weakness 1, 3. I think adding these experimental results will make this paper more insightful.

**Ethical Concerns:**

["NO or VERY MINOR ethics concerns only"]

**Final Justification:**

After reading the rebuttal, I keep my score.

**Limitations:**

Yes.

**Paper Formatting Concerns:**

No.

**Quality:**

3

**Strengths And Weaknesses:**

Pros:
1. The organization of this paper is very clear, so readers can understand the paper easily.
2. The idea of this paper is reasonable. I am happy to see that a new loss is introduced to logit distillation, since loss really plays a crucial role in logit distillation.
3. The experiment results cover various network architectures, including ViT. The improvements of the proposed method is clear and convincing.
4. The authors have discussions on the limitations of the proposed method. I think it is helpful to the research community.
5. They also provide the pseudo code and detailed theoretical analysis of the proposed method.

Cons:
1. I am curious about the comparison results with more methods, eg. DKD (CVPR2022) [1].

[1] Decoupled Knowledge Distillation

2. The authors can reorganize the experiment section, moving key comparison results forward, and put tables of baseline methods behind them.

3. Now they only provide the experiment results in classification tasks. I think the current results are enough for a NeurIPS paper, but I am still curious about the method performance on other tasks, eg. object detection.

---

> ### Author Rebuttal · Authors · 2025-07-31
>
> * We sincerely thank Reviewers fk7Y, Bcg7, ySdm, and x7Lh, as well as the Area and Senior Area Chairs, for their thorough and constructive feedback on PLD.
> * We are pleased that the reviewers recognize our method’s strong theoretical foundation (fk7Y, Bcg7), its efficiency and practical significance (fk7Y), the clarity of our derivations and writing (Bcg7, ySdm), and the robustness of our empirical evaluation (Bcg7, ySdm).
> * We greatly appreciate the suggestions to broaden our baseline comparisons, quantify runtime overhead, extend experiments to other tasks, and polish the manuscript’s readability. We will incorporate all additional experiments (CIFAR-100 baselines, COCO detection and runtime comparisons) into the revised manuscript.
> * We also thank the reviewers for highlighting minor typos and stylistic improvements, which we have addressed in the updated draft.
>
> Thank you Reviewer ySdm for the thoughtful and encouraging feedback. We’re pleased you found the paper’s organization clear, the motivation sound, and the empirical gains convincing across architectures including ViT. Your pointer to DKD (CVPR 2022) helped us broaden our baseline set, and your suggestion to reorganize the experimental section has improved clarity and flow. Without these suggestions it would have been very difficult to coordinate and execute so many additional experiments in the limited time available. In the revised manuscript we will (1) include DKD among the CIFAR-100 baselines and present key results up front with supporting tables moved to follow, (2) extend PLD to object detection on COCO to demonstrate its applicability beyond classification, and (3) refine the experimental layout to highlight these comparisons. We hope these additions further strengthen your confidence in our work.
>
>
> **Inclusion of additional KD baselines and tasks: CIFAR‑100 Baselines and Object Detection**
>
> We included DIST (NeurIPS 2022) as a baseline and structured our ImageNet-1K experiments to compare models horizontally across architectures and scales; to further strengthen the empirical comparison we added CIFAR-100 results against other recent KD methods, all trained under the same controlled hyperparameters: AdamW (β₁=0.9, β₂=0.999), learning rate $=0.001$, weight decay $=0.5$, batch size $=128$, for 250 epochs with a cosine-annealing schedule and a 10-epoch linear warm-up. Results are reported as mean $\pm$ std over three runs. Following prior work (e.g., KD and DIST) we set the temperature $\tau=4$ where applicable.
>
> Table A: CIFAR‑100 performance under identical hyperparameters
>
> | Model (Teacher→Student) | CE | PLD | KD | DIST | DKD | AT | FitNet | PKT | RKD | SP | VID |
> | --- | --- | --- | --- | --- | --- | --- | --- | --- | --- | --- | --- |
> | ResNet32x4ResNet8x4 | 71.56 ± 0.16 | **73.97 ± 0.05** | 72.11 ± 0.10 | 73.21 ± 0.10 | 73.08 ± 0.27 | 72.51 ± 0.37 | 72.59 ± 0.30 | 72.86 ± 0.17 | 71.45 ± 0.41 | 72.45 ± 0.08 | 71.70 ± 0.32 |
> | ResNet50 MobileNetV2 | 64.42 ± 0.67 | 68.16 ± 0.38 | 66.61 ± 1.16 | 66.98 ± 0.74 | **69.51 ± 0.42** | 52.28 ± 0.84 | 62.77 ± 0.05 | 66.08 ± 0.11 | 65.16 ± 0.21 | 65.30 ± 0.29 | 64.14 ± 0.82 |
> | resnet56resnet20 | 67.57 ± 0.25 | **70.28 ± 0.21** | 69.45 ± 0.24 | 69.66 ± 0.49 | 67.80 ± 0.03 | 68.42 ± 0.16 | 66.75 ± 0.27 | 68.66 ± 0.20 | 68.53 ± 0.18 | 69.36 ± 0.27 | 68.52 ± 0.12 |
> | WideResNet‑40‑2WideResNet‑40‑1 | 70.26 ± 0.24 | **73.50 ± 0.3** | 72.44 ± 0.36 | 72.74 ± 0.33 | 73.22 ± 0.38 | 71.99 ± 0.34 | 70.86 ± 0.33 | 71.43 ± 0.16 | 11.32 ± 1.84 | 73.49 ± 0.20 | 70.61 ± 0.39 |
>
> As discussed in our Conclusion (Sec 6), PLD naturally extends to any task where student and teacher share the same output logits, including detection or segmentation. To validate this, we distilled Faster R‑CNN detectors on COCO for two teacher→student pairs using DKD  and PLD:
>
> **Table B: Object detection performance (COCO)**
>
> | Teacher → Student | Method | AP | AP₅₀ | AP₇₅ | AP_s | AP_m | AP_l |
> | --- | --- | --- | --- | --- | --- | --- | --- |
> | ResNet‑50 → MobileNetV2‑FPN | DKD | **29.24** | **50.42** | 29.87 | **16.15** | **31.11** | 38.07 |
> | ResNet‑50 → MobileNetV2‑FPN | PLD | 28.91 | 49.63 | **29.88** | 16.05 | 30.41 | **38.59** |
> | ResNet‑101 → ResNet‑18‑FPN | DKD | 32.11 | 53.43 | 33.9 | 18.46 | 34.43 | 41.5 |
> | ResNet‑101 → ResNet‑18‑FPN | PLD | **32.47** | **53.83** | **34.17** | **18.5** | **34.97** | **42.12** |
> | ResNet‑101 → ResNet‑50‑FPN | DKD | 36.54 | **58.57** | 39.46 | **21.74** | **39.8** | **47.31** |
> | ResNet‑101 → ResNet‑50‑FPN | PLD | **36.6** | 58.28 | **39.58** | 21.37 | 39.76 | 47.24 |
>
> **Baselines & References**
>
> 1. **KD**: Hinton, G., Vinyals, O., & Dean, J. (2015). *Distilling the Knowledge in a Neural Network*. arXiv:1503.02531
> 2. **DIST**: Huang, T., You, S., Wang, F., Qian, C., & Xu, C. (2022). *Knowledge Distillation from A Stronger Teacher (DIST)*. arXiv:2205.10536
> 3. **DKD**: Chen, Z., Liu, W., & Tao, D. (2022). *Decoupled Knowledge Distillation*. arXiv:2203.08679
> 4. **AT**: Zagoruyko, S., & Komodakis, N. (2016). *Paying More Attention to Attention*. arXiv:1612.03928
> 5. **FitNet**: Romero, A., Ballas, N., Kahou, S. E., Chassang, A., Gatta, C., & Bengio, Y. (2015). *FitNets: Hints for Thin Deep Nets*. arXiv:1412.6550
> 6. **PKT**: Passalis, N., & Tefas, A. (2018). *Learning Deep Representations with Probabilistic Knowledge Transfer*. arXiv:1803.10837
> 7. **RKD**: Park, W., Kim, D., Lu, Y., & Cho, M. (2019). *Relational Knowledge Distillation*. arXiv:1904.05068
> 8. **SP**: Tung, F., & Mori, G. (2019). *Similarity‑Preserving Knowledge Distillation*. arXiv:1907.09682
> 9. **VID**: Ahn, S., Hu, S. X., Damianou, A., Lawrence, N. D., & Dai, Z. (2019). *Variational Information Distillation for Knowledge Transfer*. arXiv:1904.05835

---

### Official Review · Reviewer_Bcg7 · 2025-06-20

**Clarity:** 3
**Significance:** 3
**Originality:** 3
**Rating:** 5
**Confidence:** 4

**Summary:**

This paper presents a novel method for knowledge distillation (KD).  The main idea is to use a new loss function that encourages the student network's rankings for each class to match the teacher's rankings for each class and also weights each class according to the teacher's softmax class probabilities.  This loss function is based on the Plackett-Luce permutation model.  The paper shows how this new loss subsumes cross-entropy loss as well as some list-based losses.  Experimental results showing the improvement over standard KD and DIST (a recent KD method) on the ImageNet-1k dataset with various teacher-student network combinations.

**Questions:**

Are repeated experiments to compute error bars really too computationally expensive?

Why not train/test on CIFAR-100 with standard teacher-student network pairings to allow comparisons across a broad set of prior KD methods?

**Ethical Concerns:**

["NO or VERY MINOR ethics concerns only"]

**Final Justification:**

After reading the other reviews and the authors' rebuttal,  I think the paper should be accepted to NeurIPS.  The authors addressed my concerns about the lack of uncertainty estimates and lack of comparison to prior work (via results on CIFAR-100) and should include these new results in the final version of the paper.

**Limitations:**

Yes

**Paper Formatting Concerns:**

A suggestion for improving readability:  the paper frequently uses dashes to separate a dependent clause but does not include spaces around the dashes.  This makes the dash look like a hyphen which can be confusing, especially since there are also a lot of hyphenated words in the paper (such as cross-entropy).  Either include spaces around the dashes or offset dependent clauses with commas or parentheses.

**Quality:**

2

**Strengths And Weaknesses:**

Strengths:

The paper is well written and clear.  The method is well-motivated and the derivations are fairly easy to follow.

The paper is theoretically sound.  It shows a clear derivation of the new loss function and explains its connection to the Plackett-Luce permutation model.  The paper also shows how the Plackett-Luce Distillation (PLD) Loss subsumes cross-entropy loss and P-ListMLE loss.

The experiments on the ImageNet-1K dataset show improvement of PLD over KD and DIST using different teacher-student combinations (including both homogenous and heterogeneous teacher-student pairs).

Weaknesses:

Unfortunately, the experiments do not include error bars from multiple runs as has been common in many previous papers on KD.  While it is true that it would be computationally expensive to repeat all of the experiments multiple times on ImageNet-1K, it is not out of the question to repeat them say 3 times each.

The method does not compare against many previous KD methods (only KD, DIST, ListMLE and p-ListMLE).  Many past papers on KD used the CIFAR-100 dataset for training and testing with some standard teacher-student network pairings (for example, see Yonglong Tian, Dilip Krishnan, and Phillip Isola, "Contrastive representation distillation", in ICLR, 2020).  Including experiments on CIFAR-100 using the same teacher-student pairings would allow comparisons to a large number of previous KD methods.

---

> ### Author Rebuttal · Authors · 2025-07-31
>
> * We sincerely thank Reviewers fk7Y, Bcg7, ySdm, and x7Lh, as well as the Area and Senior Area Chairs, for their thorough and constructive feedback on PLD.
> * We are pleased that the reviewers recognize our method’s strong theoretical foundation (fk7Y, Bcg7), its efficiency and practical significance (fk7Y), the clarity of our derivations and writing (Bcg7, ySdm), and the robustness of our empirical evaluation (Bcg7, ySdm).
> * We greatly appreciate the suggestions to broaden our baseline comparisons, quantify runtime overhead, extend experiments to other tasks, and polish the manuscript’s readability. We will incorporate all additional experiments (CIFAR-100 baselines, COCO detection and runtime comparisons) into the revised manuscript.
> * We also thank the reviewers for highlighting minor typos and stylistic improvements, which we have addressed in the updated draft.
>
> Thank you Reviewer Bcg7 for the detailed and constructive feedback. We are glad you found the paper well written, clear, theoretically sound, and that the improvements of PLD over KD and DIST across teacher-student pairs are convincing. We also appreciate your pointer to relevant prior work such as Contrastive Representation Distillation (Tian et al., ICLR 2020), which helped guide our expansion of baselines and the design of additional CIFAR-100 experiments. Without these suggestions it would have been very difficult to coordinate and execute so many additional experiments in the limited time available. Below we (1) expand the empirical comparison with additional CIFAR-100 baselines, including uncertainty estimates from three runs to provide error bars (2) broaden the set of KD methods evaluated to strengthen the comparison and (3) demonstrate extensions to object detection. We hope these additions and clarifications address your concerns, and we would be grateful for your reconsideration of the score.
>
> **Inclusion of additional KD baselines and tasks: CIFAR‑100 Baselines and Object Detection**
>
>
> We included DIST (NeurIPS 2022) as a baseline and structured our ImageNet-1K experiments to compare models horizontally across architectures and scales; to further strengthen the empirical comparison we added CIFAR-100 results against other recent KD methods, all trained under the same controlled hyperparameters: AdamW (β₁=0.9, β₂=0.999), learning rate $=0.001$, weight decay $=0.5$, batch size $=128$, for 250 epochs with a cosine-annealing schedule and a 10-epoch linear warm-up. Results are reported as mean $\pm$ std over three runs. Following prior work (e.g., KD and DIST) we set the temperature $\tau=4$ where applicable.
>
> Table A: CIFAR‑100 performance under identical hyperparameters
>
> | Model (Teacher→Student) | CE | PLD | KD | DIST | DKD | AT | FitNet | PKT | RKD | SP | VID |
> | --- | --- | --- | --- | --- | --- | --- | --- | --- | --- | --- | --- |
> | ResNet32x4ResNet8x4 | 71.56 ± 0.16 | **73.97 ± 0.05** | 72.11 ± 0.10 | 73.21 ± 0.10 | 73.08 ± 0.27 | 72.51 ± 0.37 | 72.59 ± 0.30 | 72.86 ± 0.17 | 71.45 ± 0.41 | 72.45 ± 0.08 | 71.70 ± 0.32 |
> | ResNet50 MobileNetV2 | 64.42 ± 0.67 | 68.16 ± 0.38 | 66.61 ± 1.16 | 66.98 ± 0.74 | **69.51 ± 0.42** | 52.28 ± 0.84 | 62.77 ± 0.05 | 66.08 ± 0.11 | 65.16 ± 0.21 | 65.30 ± 0.29 | 64.14 ± 0.82 |
> | resnet56resnet20 | 67.57 ± 0.25 | **70.28 ± 0.21** | 69.45 ± 0.24 | 69.66 ± 0.49 | 67.80 ± 0.03 | 68.42 ± 0.16 | 66.75 ± 0.27 | 68.66 ± 0.20 | 68.53 ± 0.18 | 69.36 ± 0.27 | 68.52 ± 0.12 |
> | WideResNet‑40‑2WideResNet‑40‑1 | 70.26 ± 0.24 | **73.50 ± 0.3** | 72.44 ± 0.36 | 72.74 ± 0.33 | 73.22 ± 0.38 | 71.99 ± 0.34 | 70.86 ± 0.33 | 71.43 ± 0.16 | 11.32 ± 1.84 | 73.49 ± 0.20 | 70.61 ± 0.39 |
>
> As discussed in our Conclusion (Sec 6), PLD naturally extends to any task where student and teacher share the same output logits, including detection or segmentation. To validate this, we distilled Faster R‑CNN detectors on COCO for two teacher→student pairs using DKD  and PLD:
>
> **Table B: Object detection performance (COCO)**
>
> | Teacher → Student | Method | AP | AP₅₀ | AP₇₅ | AP_s | AP_m | AP_l |
> | --- | --- | --- | --- | --- | --- | --- | --- |
> | ResNet‑50 → MobileNetV2‑FPN | DKD | **29.24** | **50.42** | 29.87 | **16.15** | **31.11** | 38.07 |
> | ResNet‑50 → MobileNetV2‑FPN | PLD | 28.91 | 49.63 | **29.88** | 16.05 | 30.41 | **38.59** |
> | ResNet‑101 → ResNet‑18‑FPN | DKD | 32.11 | 53.43 | 33.9 | 18.46 | 34.43 | 41.5 |
> | ResNet‑101 → ResNet‑18‑FPN | PLD | **32.47** | **53.83** | **34.17** | **18.5** | **34.97** | **42.12** |
> | ResNet‑101 → ResNet‑50‑FPN | DKD | 36.54 | **58.57** | 39.46 | **21.74** | **39.8** | **47.31** |
> | ResNet‑101 → ResNet‑50‑FPN | PLD | **36.6** | 58.28 | **39.58** | 21.37 | 39.76 | 47.24 |
>
> **Baselines & References**
>
> 1. **KD**: Hinton, G., Vinyals, O., & Dean, J. (2015). *Distilling the Knowledge in a Neural Network*. arXiv:1503.02531
> 2. **DIST**: Huang, T., You, S., Wang, F., Qian, C., & Xu, C. (2022). *Knowledge Distillation from A Stronger Teacher (DIST)*. arXiv:2205.10536
> 3. **DKD**: Chen, Z., Liu, W., & Tao, D. (2022). *Decoupled Knowledge Distillation*. arXiv:2203.08679
> 4. **AT**: Zagoruyko, S., & Komodakis, N. (2016). *Paying More Attention to Attention*. arXiv:1612.03928
> 5. **FitNet**: Romero, A., Ballas, N., Kahou, S. E., Chassang, A., Gatta, C., & Bengio, Y. (2015). *FitNets: Hints for Thin Deep Nets*. arXiv:1412.6550
> 6. **PKT**: Passalis, N., & Tefas, A. (2018). *Learning Deep Representations with Probabilistic Knowledge Transfer*. arXiv:1803.10837
> 7. **RKD**: Park, W., Kim, D., Lu, Y., & Cho, M. (2019). *Relational Knowledge Distillation*. arXiv:1904.05068
> 8. **SP**: Tung, F., & Mori, G. (2019). *Similarity‑Preserving Knowledge Distillation*. arXiv:1907.09682
> 9. **VID**: Ahn, S., Hu, S. X., Damianou, A., Lawrence, N. D., & Dai, Z. (2019). *Variational Information Distillation for Knowledge Transfer*. arXiv:1904.05835

---

> > ### Comment · Reviewer_Bcg7 · 2025-08-04
> >
> > The authors did a good job of responding to my concerns about the lack of uncertainty estimates and lack of comparison to prior work (via results on CIFAR-100).  I think the paper should be accepted to NeurIPS.

---

> > > ### Author Response · Authors · 2025-08-05
> > >
> > > Thank you for your time and thoughtful feedback. We appreciate your constructive insights and your support in guiding our revisions.

---

### Official Review · Reviewer_fk7Y · 2025-06-27

**Clarity:** 2
**Significance:** 2
**Originality:** 3
**Rating:** 4
**Confidence:** 4

**Summary:**

This paper proposes Plackett-Luce Distillation (PLD), a novel list-wise, choice-theoretic knowledge distillation loss that leverages the Plackett-Luce permutation model to enforce a teacher-optimal ranking of classes. By interpreting teacher logits as worth scores and weighting each selection step by the teacher's confidence, PLD provides a convex, translation-invariant surrogate that unifies several losses. Experiments on ImageNet-1K demonstrate consistent accuracy improvements over standard KL-based distillation (KD) and correlation-based distillation (DIST) across various student-teacher architectures.

**Questions:**

See Weaknesses.

**Ethical Concerns:**

["NO or VERY MINOR ethics concerns only"]

**Final Justification:**

Thank you for your constructive and insightful comments on my manuscript. I have carefully considered your feedback, and I am pleased to inform you that your responses have addressed the majority of my concerns. Your suggestions have significantly clarified several aspects of the paper, and I now feel more confident in its contribution to the field.

As a result, I have revised my assessment and have decided to increase the score accordingly.

**Limitations:**

1.The proposed method may require a lot of additional training time.
2.The experiments are limited to image classification. Can the proposed method works on other tasks?

**Paper Formatting Concerns:**

None.

**Quality:**

3

**Strengths And Weaknesses:**

Strengths
1.The paper introduces a well-founded framework based on choice theory and the Plackett-Luce model.
2.The method is convex and differentiable, with closed-form gradients, making it efficient and amenable to gradient-based optimization.
3.The proposed method eliminates the need for manual tuning of weights between the cross-entropy and distillation terms.

Weaknesses
1.Converting the teacher's outputs into a ranked list and then using the Plackett-Luce model to weight labels based on logits seems to me like a simple remapping of the original soft labels. I don't think this approach extracts significantly more useful information from the soft labels.
2.The comparison methods in this paper are too limited. Please refer to other knowledge distillation (KD) papers and include more baseline algorithms for comparison.
3.This paper introduces the Plackett-Luce model and requires computation of closed-form gradients, which incurs significant additional training cost. Please provide a comparison of training time to quantify this overhead.
4.Potential applications beyond image classification (e.g., NLP, Image Segmentation) are not explored experimentally.
5.This paper is difficult to understand.

---

> ### Author Rebuttal · Authors · 2025-07-31
>
> * We sincerely thank Reviewers fk7Y, Bcg7, ySdm, and x7Lh, as well as the Area and Senior Area Chairs, for their thorough and constructive feedback on PLD.
> * We are pleased that the reviewers recognize our method’s strong theoretical foundation (fk7Y, Bcg7), its efficiency and practical significance (fk7Y), the clarity of our derivations and writing (Bcg7, ySdm), and the robustness of our empirical evaluation (Bcg7, ySdm).
> * We greatly appreciate the suggestions to broaden our baseline comparisons, quantify runtime overhead, extend experiments to other tasks, and polish the manuscript’s readability. We will incorporate all additional experiments (CIFAR-100 baselines, COCO detection and runtime comparisons) into the revised manuscript.
> * We also thank the reviewers for highlighting minor typos and stylistic improvements, which we have addressed in the updated draft.
>
> Thank you Reviewer fk7Y for the detailed and constructive feedback. We’re glad the reviewers (including fk7Y and others) recognize PLD’s choice-theoretic foundation, its convex differentiable surrogate with closed-form gradients, and its elimination of manual tuning between cross-entropy and distillation. Below we (1) show how PLD transfers richer intra-class information beyond soft-label remapping, (2) present expanded CIFAR-100 baselines under identical hyperparameters, (3) provide empirical runtime analyses, (4) demonstrate extensions to object detection, and (5) describe improvements to clarity and consistency in the manuscript.
> We hope these additions and clarifications address your concerns, and we would be grateful for your reconsideration of the score.
>
> **Q1. Is converting the teacher’s outputs into a ranked list and then using the Plackett–Luce model to weight labels based on logits merely a remapping of the original soft labels, or does it extract substantially more useful information?**
>
> Answer:  PLD extracts richer intra-class information than standard KD by supervising the full teacher permutation with confidence-weighted list-wise ranking, instead of just remapping soft labels.
>
> In knowledge distillation we add extra constraints to the loss to extract richer information from the teacher. Cross-entropy alone only pushes the student toward the top class, whereas distillation terms transfer subtler inter-class relations. PLD uses *weighted list-wise ranking*: given a teacher ranking $a > b > c > d > e$, supervising this single permutation not only enforces $a$ as the top choice but also embeds every relative preference (e.g., $a>b>c$, $b>c>d$, etc.). Furthermore, each selection step is weighted by the teacher’s softmax confidence (so $\alpha_1 = q^T_a \gg \alpha_5 = q^T_e$), meaning PLD emphasizes high-confidence distinctions like $a>b$ over low-confidence ones like $d>e$, all within one forward pass.
>
> By contrast, standard KD (CE + KL) depends on softmax normalization: when the teacher is highly confident, the target is near one-hot and carries little structure, so the KL term must be softened via temperature (Hinton et al., 2015) just to expose any ordering. PLD instead directly minimizes the negative log-likelihood of the *entire* teacher-optimal permutation $\pi^* = (y, \text{argsort}(t)\setminus\{y\})$, aligning all ordering relations in a single convex loss (Sec. 4.1, L183–L184), and uses confidence weights $\alpha_k = q^T_{\pi^*_k}$ to prioritize the strongest signals (Sec. 4.2). This yields substantially richer intra-class information than mere soft-label remapping.
>
> **Q2: Inclusion of additional KD baselines.**
>
> Answer: We included DIST (NeurIPS 2022) as a baseline and structured our ImageNet-1K experiments to compare models horizontally across architectures and scales; to further strengthen the empirical comparison we added CIFAR-100 results against other recent KD methods, all trained under the same controlled hyperparameters: AdamW (β₁=0.9, β₂=0.999), learning rate $=0.001$, weight decay $=0.5$, batch size $=128$, for 250 epochs with a cosine-annealing schedule and a 10-epoch linear warm-up. Results are reported as mean $\pm$ std over three runs. Following prior work (e.g., KD and DIST) we set the temperature $\tau=4$ where applicable.
>
> Table A: CIFAR‑100 performance under identical hyperparameters
>
> | Model (Teacher→Student) | CE | PLD | KD | DIST | DKD | AT | FitNet | PKT | RKD | SP | VID |
> | --- | --- | --- | --- | --- | --- | --- | --- | --- | --- | --- | --- |
> | ResNet32x4ResNet8x4 | 71.56 ± 0.16 | **73.97 ± 0.05** | 72.11 ± 0.10 | 73.21 ± 0.10 | 73.08 ± 0.27 | 72.51 ± 0.37 | 72.59 ± 0.30 | 72.86 ± 0.17 | 71.45 ± 0.41 | 72.45 ± 0.08 | 71.70 ± 0.32 |
> | ResNet50 MobileNetV2 | 64.42 ± 0.67 | 68.16 ± 0.38 | 66.61 ± 1.16 | 66.98 ± 0.74 | **69.51 ± 0.42** | 52.28 ± 0.84 | 62.77 ± 0.05 | 66.08 ± 0.11 | 65.16 ± 0.21 | 65.30 ± 0.29 | 64.14 ± 0.82 |
> | resnet56resnet20 | 67.57 ± 0.25 | **70.28 ± 0.21** | 69.45 ± 0.24 | 69.66 ± 0.49 | 67.80 ± 0.03 | 68.42 ± 0.16 | 66.75 ± 0.27 | 68.66 ± 0.20 | 68.53 ± 0.18 | 69.36 ± 0.27 | 68.52 ± 0.12 |
> | WideResNet‑40‑2WideResNet‑40‑1 | 70.26 ± 0.24 | **73.50 ± 0.3** | 72.44 ± 0.36 | 72.74 ± 0.33 | 73.22 ± 0.38 | 71.99 ± 0.34 | 70.86 ± 0.33 | 71.43 ± 0.16 | 11.32 ± 1.84 | 73.49 ± 0.20 | 70.61 ± 0.39 |
>
>
>
> **Q3. Does introducing the Plackett–Luce model with its closed-form gradients incur significant additional training cost?**
>
> Answer: PLD’s runtime is on par with state-of-the-art KD methods while yielding the high efficiency. PLD enforces only a single teacher‑optimal permutation (O(C log C) via sorting) rather than summing over all C! rankings—just as ListMLE  and P‑ListMLE do—making it computationally feasible and directly comparable to other state‑of‑the‑art distillation methods. Below is the run time of Table A’s methods:
>
> **Table B: Total training time (minutes)**
>
> | Runtime | CE | PLD | KD | DIST | DKD | AT | FitNet | PKT | RKD | SP | VID |
> | --- | --- | --- | --- | --- | --- | --- | --- | --- | --- | --- | --- |
> | ResNet32x4ResNet8x4 | 32.06 | 62.84 | 62.18 | 62.24 | 62.46 | 65.39 | 63.96 | 62.53 | 68.36 | 62.73 | 78.40 |
> | ResNet50 MobileNetV2 | 23.62 | 102.59 | 102.23 | 102.49 | 102.54 | 108.17 | 113.18 | 102.28 | 128.65 | 102.78 | 132.06 |
> | resnet56resnet20 | 22.97 | 33.42 | 33.57 | 33.81 | 33.46 | 34.55 | 33.30 | 33.89 | 36.21 | 33.67 | 36.57 |
> | WideResNet‑40‑2WideResNet‑40‑1 | 27.67 | 33.1 | 37.56 | 36.73 | 34.27 | 36.77 | 38.05 | 36.84 | 37.62 | 37.01 | 37.35 |
> Despite its list‑wise nature, PLD’s runtime is on par with KD, DIST, and DKD, and substantially faster than RKD or VID. Furthermore, we found PLD delivers the high accuracy gain over time
>
> **Table C: Accuracy gain over time (ΔTop‑1 / runtime)**
>
> | gain | CE | PLD | KD | DIST | DKD | AT | FitNet | PKT | RKD | SP | VID |
> | --- | --- | --- | --- | --- | --- | --- | --- | --- | --- | --- | --- |
> | ResNet32x4ResNet8x4 | 0 | **0.038** | 0.009 | 0.027 | 0.024 | 0.015 | 0.016 | 0.021 | -0.002 | 0.014 | 0.002 |
> | ResNet50 MobileNetV2 | 0 | 0.036 | 0.021 | 0.025 | **0.05** | -0.112 | -0.015 | 0.016 | 0.006 | 0.009 | -0.002 |
> | resnet56resnet20 | 0 | **0.081** | 0.056 | 0.062 | 0.007 | 0.025 | -0.025 | 0.032 | 0.027 | 0.053 | 0.026 |
> | WideResNet‑40‑2WideResNet‑40‑1 | 0 | **0.0979** | 0.0581 | 0.0676 | 0.0864 | 0.0471 | 0.0158 | 0.0318 | −1.5670 | 0.0873 | 0.0094 |
>
> **Q4. Why are potential applications beyond image classification (e.g., NLP, image segmentation) not explored experimentally?**
>
> Answer: As discussed in our Conclusion (Sec 6), PLD naturally extends to any task where student and teacher share the same output logits, including detection or segmentation. To validate this, we distilled Faster R‑CNN detectors on COCO for two teacher→student pairs using DKD  and PLD:
>
> **Table D: Object detection performance (COCO)**
>
> | Teacher → Student | Method | AP | AP₅₀ | AP₇₅ | AP_s | AP_m | AP_l |
> | --- | --- | --- | --- | --- | --- | --- | --- |
> | ResNet‑50 → MobileNetV2‑FPN | DKD | **29.24** | **50.42** | 29.87 | **16.15** | **31.11** | 38.07 |
> | ResNet‑50 → MobileNetV2‑FPN | PLD | 28.91 | 49.63 | **29.88** | 16.05 | 30.41 | **38.59** |
> | ResNet‑101 → ResNet‑18‑FPN | DKD | 32.11 | 53.43 | 33.9 | 18.46 | 34.43 | 41.5 |
> | ResNet‑101 → ResNet‑18‑FPN | PLD | **32.47** | **53.83** | **34.17** | **18.5** | **34.97** | **42.12** |
> | ResNet‑101 → ResNet‑50‑FPN | DKD | 36.54 | **58.57** | 39.46 | **21.74** | **39.8** | **47.31** |
> | ResNet‑101 → ResNet‑50‑FPN | PLD | **36.6** | 58.28 | **39.58** | 21.37 | 39.76 | 47.24 |
>
> Given PLD’s competitive performance, we hope other researchers will adopt it for additional domains such as NLP, as discussed in Section 6 (L303).
>
> **Q5. This paper is difficult to understand.**
>
> Answer: While we appreciate this feedback, several other reviewers found the paper clear and accessible. For example, Reviewer Bcg7 writes, “The paper is well written and clear. The method is well-motivated and the derivations are fairly easy to follow.” and Reviewer ySdm notes, “The organization of this paper is very clear, so readers can understand the paper easily.” Nonetheless, we’ve further smoothed the narrative flow by breaking down long, dash‑connected sentences into shorter, standalone sentences and adding brief transitional phrases, making the text easier to follow. We further appreciate your detailed suggestions for improving the writing and will incorporate them in the revised manuscript.
>
> **Baselines & References**
>
> 1. **KD**:     arXiv:1503.02531
> 2. **DIST**:  arXiv:2205.10536
> 3. **DKD**:   arXiv:2203.08679
> 4. **AT**:      arXiv:1612.03928
> 5. **FitNet**:arXiv:1412.6550
> 6. **PKT**:   arXiv:1803.10837
> 7. **RKD**:  arXiv:1904.05068
> 8. **SP**:     arXiv:1907.09682
> 9. **VID**:    arXiv:1904.05835

---

> ### Author Response · Authors · 2025-08-06
>
> Dear Reviewer fk7Y,
>
> thank you once again for taking the time to review our paper, **PLD: A Choice-Theoretic List-Wise Knowledge Distillation**. We have submitted our rebuttal and provided clarifications addressing your comments, and as the rebuttal period is progressing, we would greatly appreciate it if you could kindly take a moment to review our response to ensure it adequately addresses your concerns. Your feedback is invaluable to us, and we sincerely hope that our clarifications can help you reconsider our work in light of the points addressed. If you have any further questions or require additional details, please do not hesitate to contact us. Thank you very much for your time and consideration.
>
> Best regards,
>
> Authors of **PLD: A Choice-Theoretic List-Wise Knowledge Distillation**

---

> ### Comment · Reviewer_fk7Y · 2025-08-09
>
> Thank you for your constructive and insightful comments on my manuscript. I have carefully considered your feedback, and I am pleased to inform you that your responses have addressed the majority of my concerns. Your suggestions have significantly clarified several aspects of the paper, and I now feel more confident in its contribution to the field.
>
> As a result, I have revised my assessment and have decided to increase the score accordingly. I believe the paper does not deserve a rating of 5 because it only includes two baselines on ImageNet-1K. Please include more comparative experimental results.

---

> > ### Author Response · Authors · 2025-08-09
> >
> > Thank you for your reconsideration of the score and for your review of our work. We are pleased to hear that our responses have addressed the majority of your concerns. We appreciate your suggestions regarding additional comparative experiments. While the limited time during the discussion period did not allow us to run further experiments on ImageNet-1K, we have provided CIFAR-100 results including many other baselines to address the request for broader comparisons.

---

### Comment · Area_Chair_mtY2 · 2025-08-04
**Please Read the Rebuttal and Discuss**

Dear Reviewers Bcg7, fk7Y. x7Lh, ySdm

The authors have submitted their rebuttal.

Please carefully review all other reviews and the authors’ responses, and engage in an open exchange with the authors.

Kindly post your initial response as early as possible within the discussion window to allow sufficient time for interaction.

Your AC

---

### Author Response · Authors · 2025-08-09

* We would like to express our sincere gratitude to the meta-reviewer, the area chair, and all the reviewers for taking the time to evaluate our paper. We truly appreciate your valuable feedback and insightful comments.

* We are pleased that the reviewers acknowledge our choice-theoretic formulation and the Plackett–Luce–derived loss for knowledge distillation (fk7Y, Bcg7, ySdm, x7Lh); that PLD offers a unified perspective on logit-based distillation (Bcg7, x7Lh, fk7Y, ySdm); and that the paper is clearly organized and well-motivated (Bcg7, ySdm). They also recognized our efforts to share pseudocode and detailed theoretical analysis.

* Regarding the issues raised by the reviewers, we greatly value these points and have addressed them in detail in our rebuttal. We are glad that most concerns appear to have been resolved. While it is unfortunate that the limited time during the discussion period did not allow us to run additional experiments on ImageNet-1K, we have provided CIFAR-100 results including many other baselines to address the request for broader comparisons.

* We thank the reviewers once again for their careful reading and for pointing out typos and minor revisions, which we will promptly correct in the final version.

---

### Decision · Program_Chairs · 2025-09-17

**Decision:**

Accept (poster)

**Comment:**

This paper introduces Plackett–Luce Distillation (PLD), a choice-theoretic, list-wise knowledge distillation loss that unifies cross-entropy and distillation terms in a convex, differentiable framework. The method is theoretically well-motivated and demonstrates consistent improvements over standard KD and DIST on ImageNet-1K.

Reviewers initially raised several concerns. Reviewer fk7Y questioned whether PLD extracted more information beyond soft-label remapping, criticized the limited baseline coverage, and flagged potential runtime overhead and unclear writing. Reviewer Bcg7 found the paper clear and theoretically sound but requested uncertainty estimates and comparisons to a broader set of KD methods. Reviewer ySdm found the paper well-organized and convincing but asked for inclusion of DKD baselines and suggested reorganization of the experimental section, while also expressing curiosity about applicability beyond classification. Reviewer x7Lh felt the writing needed polish, the motivation was not clearly argued, and empirical comparisons were too narrow.

The rebuttal addressed these points: the authors provided expanded CIFAR-100 baselines including DKD and other methods, reported runtime analysis showing PLD comparable to KD and DIST, added COCO detection experiments to demonstrate generalization beyond classification, and revised the manuscript for clarity. These additions persuaded most reviewers. Reviewer Bcg7 and ySdm endorsed acceptance, satisfied with the new results. Reviewer x7Lh acknowledged the clarifications and raised the score to borderline accept. Reviewer fk7Y recognized that many concerns were addressed and also raised the score.

Overall, the consensus is that PLD is a technically solid and good contribution to knowledge distillation. With two reviewers recommending acceptance and two moving to borderline accept after rebuttal, the balance of opinion is that the paper merits acceptance.